# Unique exploration method of electronic component failure time and yield strengths of circular tubes under complete flexible model

**Riffat Jabeen**[1]*, **Mashhood Ahmad**[1], **Azam Zaka**[2], **M. Nagy**[3], **Hazem Al-Mofleh**[4]

**1** Department of Statistics, COMSATS University Islamabad, Lahore Campus, Lahore, Pakistan,
**2** Department of Statistics, Government Graduate College of Science, Lahore, Pakistan, **3** Department of Statistics and Operations Research, College of Science, King Saud University, Riyadh, Saudi Arabia,
**4** Department of Mathematical Sciences, Ball State University, Muncie, Indiana, United States of America

* riffat.jabeen79@gmail.com

**Data Availability Statement:** All relevant data are within the article.

## Abstract

The aim of this study is to develop new exponential weighted moving average control charts based on a flexible model. These control charts created through least square and weighted least square estimators of the shape parameter of the new Kumaraswamy Pareto distribution. Exponential weighted moving average control charts based on least square and weighted least square estimators are compared for checking the performance of control charts. The results were not only explored through numerical values but also explored through half a dozen plots. The numerical results and plots exposed that the exponential weighted moving average control chart based on weighted least square estimator has better performance than the other proposed chart. Some key findings are discussed which are obtained from the comparative analysis of EWMA control charts. The simulation study of proposed charts is also reported in detail. The two data sets further demonstrate the effectiveness of the proposed charts. The reported results, for real data sets, are not only displayed in normal plots but also displayed in three-dimension plots. We recommend that the proposed method can be adapted for different types of distributions, and also suggest some future research directions. The concluding remarks are reported at the end of this manuscript.

## 1. Introduction

The purpose of this study is to develop new exponential weighted moving average (EWMA) control charts to monitor the shape parameter based on different estimators of new Kumaraswamy Pareto (NKP) distribution [1]. The NKP distribution can fit on every type of data whether it is positive skewed, negative skewed, reverse J shape, J shape, symmetric or u shape. The new EWMA control charts based on least square and weighted least square estimator of shape parameter of the (NKP) distribution. [2, 3] supported the statements made about the NKP distribution.

The least square and weighted least square are regression based estimation methods of parameters that are originally proposed by [4] to estimate parameters of beta distribution. Let $Y_1, Y_2, \ldots, Y_n$ is a random sample with size n from the distribution function $G(\cdot)$. Assume that

**Funding:** This research was conducted under a project titled "Researchers Supporting Project", funded by King Saud University, Riyadh, Saudi Arabia under grant number (RSPD2024R969).

**Competing interests:** The authors have declared that no competing interests exist.

$Y_{(i)}$, is ordered statistics, where $i = 1,2,\ldots n$. The proposed method uses the distribution function $G(Y_{(i)})$. The mean and variance of the $G(Y_{(i)})$ are given below

$$E\left(G(Y_{(i)})\right) = \frac{i}{n+1}$$

and

$$v\left(G(Y_{(i)})\right) = \frac{i(n-i+1)}{(n+1)^2(n+2)}$$

The least square estimator (LSE) of the unknown parameters obtained by minimizing the function

$$LSE = \sum_{i=1}^{n}\left[G\left(Y_{(i)}\right) - \left(\frac{i}{n+1}\right)\right]^2, \ i = 1,2,\ldots n$$

with respect to unknown parameters.

The weighted least square estimator (WLSE) of the unknown parameters can be obtained by minimizing:

$$WLSE = \sum_{i=1}^{n} w\left[G\left(Y_{(i)}\right) - \left(\frac{i}{n+1}\right)\right]^2, \ \ i = 1,2,\ldots n,$$

with respect to unknown parameters, where $w = \frac{1}{\left(v(G(Y_{(n)}))\right)} = \frac{1}{\left(\frac{i(n-i+1)}{(n+1)^2(n+2)}\right)} = \frac{(n+1)^2(n+2)}{i(n-i+1)}$

Control charts are broadly used as the most convenient tool in checking product quality in agricultural and industrial production processes. The EWMA chart was proposed for the first time by [5]. The Weibull scale parameter is based on type I censored data using a modified exponential weighted moving average (MEWMA) control chart introduced by [6]. The properties of the exponential EWMA (EEWMA) control chart with parameter estimation are discussed in [7]. Additionally, the effect of parameter estimation on the performance measures of the EEWMA was investigated by [7]. A new EWMA control chart is discussed in [8]. This new chart is based on log transformation of the sample variance. A comparative study was reported to check the performance of this new chart with some previous usual charts. In [9], the new EWMA chart was tested based on sample range of logarithms of data and an unbiased estimator. The EWMA and adaptive EWMA are discussed in [10]. This EWMA chart is used to monitor both parameters such as shape and scale parameter of Weibull distribution. Some new EWMA charts for monitoring the mean of censored Weibull lifetimes reported in [11]. In [12], EWMA control chart was developed based on the shape parameter of Weibull process. Moreover, the performance of the proposed control chart is checked by comparing the values of average run length (ARL) with existing control chart. The control charts for monitoring the Weibull shape parameter based on type-II censored sample were investigated by [13]. The modified control charts for monitoring the shape parameter of weighted power function distribution under classical estimator was discussed in [14]. Different control charts are proposed. The simulation study and application on real data sets of the proposed charts are also reported. [15] discussed the new EWMA chart for simultaneously monitoring the parameters of a shifted exponential distribution. [16] investigated the inertial properties of EWMA control charts. [17] obtained the generally weighted moving average control chart for monitoring the process mean of autocorrelated observations. Some other new EWMA charts were also investigated in [18–21].

The R statistical programming software is used to obtain results in the simulation study and the applications. The rest of the manuscript is prepared as: The new Kumaraswamy-Pareto (NKP) distribution is discussed in section 2. In Section 3, EWMA control charts of NKP distribution based on LSE and WLSE are proposed. In Section 4, the applications of proposed control charts are provided on two data sets. In Section 5, we discussed more about the proposed method and how it can be adapted for different types of data distributions. The future research directions are specified in Section 6 and the concluding remarks are given in Section 7.

## 2. New Kumaraswamy-Pareto (NKP) distribution

The CDF and the PDF of NKP distribution are, respectively, given in [1] as follows

$$G_{\text{NKP}}(y) = 1 - \left[1 - \left(\frac{y}{\alpha}\right)^{\gamma}\right]^{\theta}, \ 0 < y < \alpha, \ \alpha, \theta, \gamma > 0 \tag{1}$$

$$g_{\text{NKP}}(y) = \gamma\theta y^{\gamma-1}\alpha^{-\gamma}\left[1 - \left(\frac{y}{\alpha}\right)^{\gamma}\right]^{\theta-1}, \ \ 0 < y < \alpha. \tag{2}$$

The studied model can fit every type of data whether it is positive skewed, negative skewed, reverse J shape, J shape, symmetric or U shape. The NKP model is a very flexible model as we can see in Figs 1 and 2. Several other flexible shapes of NKP model can be seen in its baseline article [1]. Besides the other parameters, there is a significant role of a shape parameter, $\alpha$, in NKP model. The shape parameter controls the shape of NKP model. By using different values for the shape parameter, several shapes can be attained.

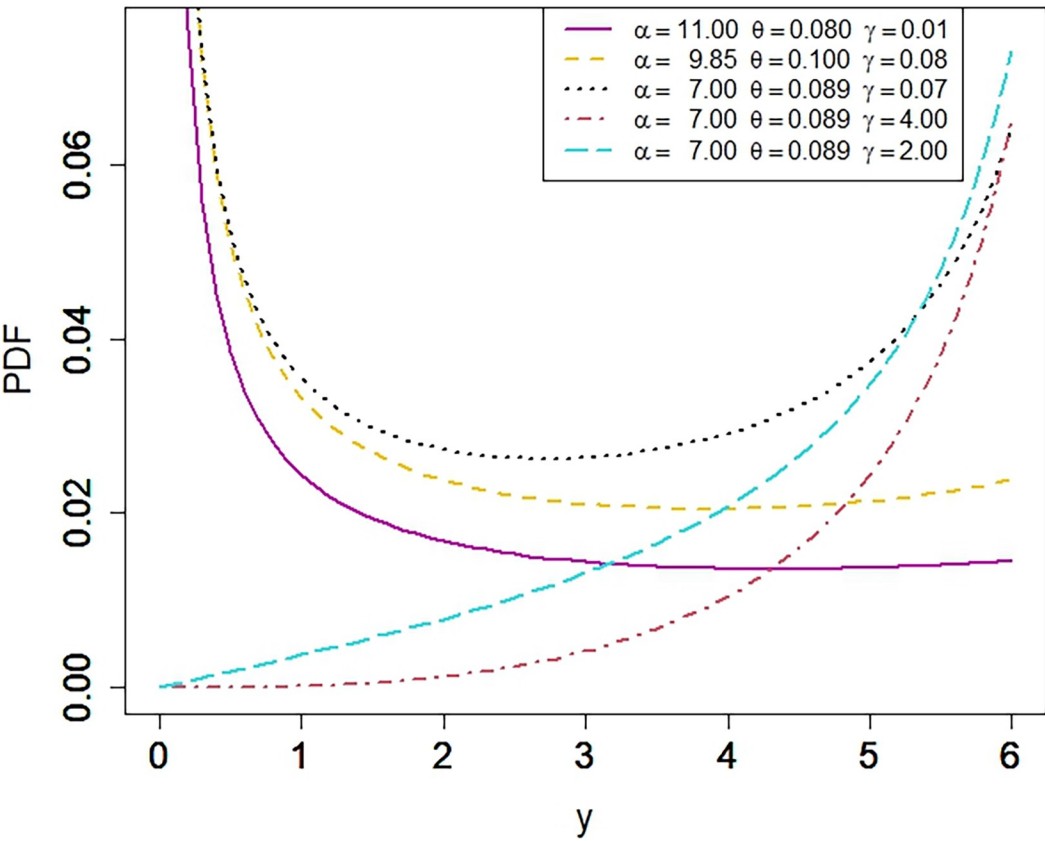

**Fig 1. Plot for the PDF of the NKP distribution.**

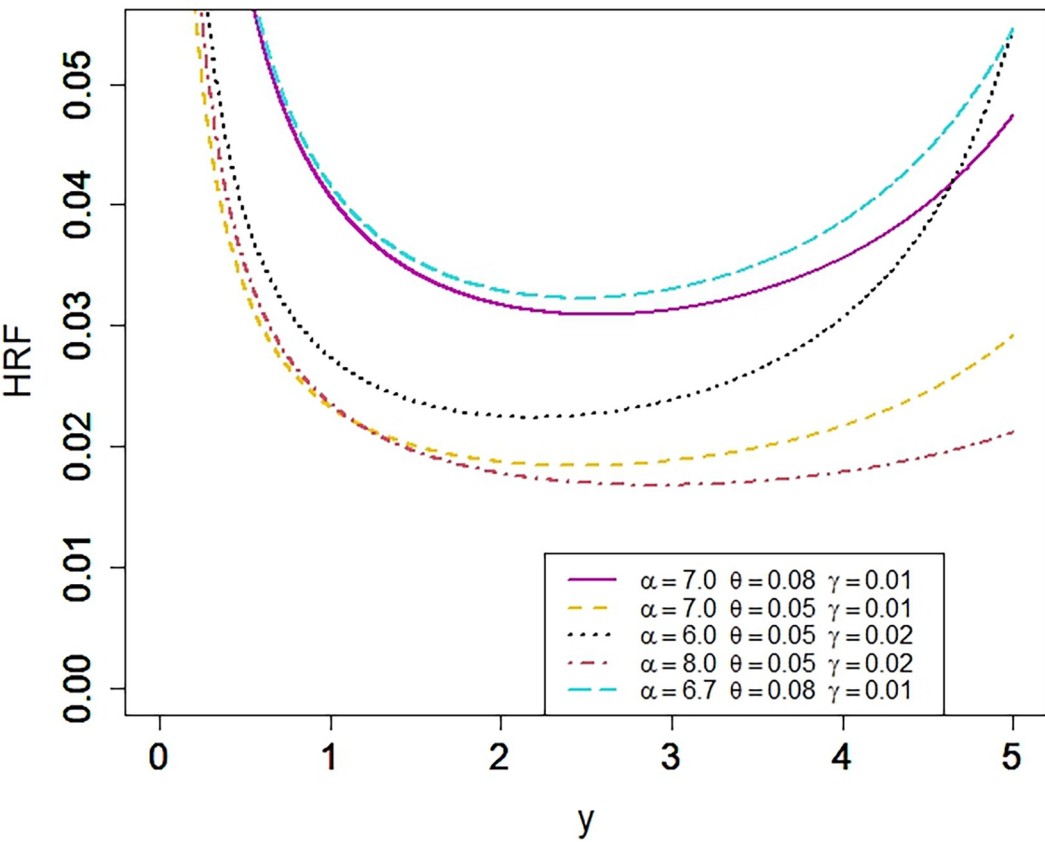

**Fig 2. Plot for the HRF of the NKP distribution.**

## 2.1. Estimation methods

The LSE and WLSE are discussed by [4]. The LSE and WLSE of NKP distribution are discussed by [1]. In this study these methods are used to develop new control charts. The R language is used for computing the results. These methods are also used in [22–25].

**2.1.1. Least-square estimators.** The LSE of the NKP parameters obtained by minimizing the function

$$LSE(\gamma, \theta, \alpha) = \sum_{i=1}^{n} \left[ G_{\mathrm{NKP}}\left(Y_{(i)}|\gamma, \theta, \alpha\right) - \left(\frac{i}{n+1}\right) \right]^2, \quad i = 1, 2, \ldots n, \tag{3}$$

with respect to $\gamma, \theta, \alpha$. The LSE are obtained by solving the following non-linear equations

$$\sum_{i=1}^{n} \left[ G_{\mathrm{NKP}}\left(Y_{(i)}|\gamma, \theta, \alpha\right) - \left(\frac{i}{n+1}\right) \right] \Delta_r\left(Y_{(i)}|\gamma, \theta, \alpha\right) = 0, \quad i = 1, 2, \ldots n; \; r = 1, 2, 3,$$

where

$$\Delta_1\left(Y_{(i)}|\gamma, \theta, \alpha\right) = \frac{\partial}{\partial \gamma}\left[G_{\mathrm{NKP}}(Y_{(i)}|\gamma, \theta, \alpha)\right],$$

$$\Delta_2\left(Y_{(i)}|\gamma, \theta, \alpha\right) = \frac{\partial}{\partial \theta}\left[G_{\mathrm{NKP}}(Y_{(i)}|\gamma, \theta, \alpha)\right],$$

$$\Delta_3\left(Y_{(i)}|\gamma, \theta, \alpha\right) = \frac{\partial}{\partial \alpha}\left[G_{\mathrm{NKP}}\left(Y_{(i)}|\gamma, \theta, \alpha\right)\right].$$

**2.1.2. Weighted least-square estimators.** The WLSE of the NKP parameters can be obtained by minimizing:

$$WLSE(\gamma, \theta, \alpha) = \sum_{i=1}^{n} w\left[G_{\mathrm{NKP}}\left(Y_{(i)}|\gamma, \theta, \alpha\right) - \left(\frac{i}{n+1}\right)\right]^2, \quad i = 1, 2, \ldots n, \tag{4}$$

with respect to $\gamma, \theta, \alpha$, where $w = \frac{1}{(v(G(Y_{(n)})))} = \frac{1}{\left(\frac{i(n-i+1)}{(n+1)^2(n+2)}\right)} = \frac{(n+1)^2(n+2)}{i(n-i+1)}$

Furthermore, the WLSE obtained by solving the non-linear equations:

$$\sum_{i=1}^{n}\left(\frac{(n+1)^2(n+2)}{i(n-i+1)}\right)\left[G_{\mathrm{NKP}}\left(Y_{(i)}|\gamma, \theta, \alpha\right) - \left(\frac{i}{n+1}\right)\right]\Delta_r\left(Y_{(i)}|\gamma, \theta, \alpha\right) = 0$$

$$i = 1, 2, \ldots n; \ r = 1, 2, 3,$$

where $\Delta_r(Y_{(i)}|\gamma, \theta, \alpha)$, for $r$ = 1,2,3 are defined before in the previous subsection.

# 3. Exponentially weighted moving average control charts (EWMA CC)

In existing exponentially weighted moving average control charts (EWMA CC), the EWMA statistics, $Z_t$, at time $t$ based on variable $Y$ is $Z_t = \lambda Y_t + (1-\lambda)Z_{t-1}$ where $Z_t$ is EWMA statistics on current time, $Z_{t-1}$ is EWMA statistics on previous time and $\lambda$ is a smoothing constant which range is $0 < \lambda \leq 1$. The small values of smoothing constant $\lambda$ tell us that the less weight is given to current value of the variable. The large values of smoothing constant $\lambda$ tell us that the more weight is given to previous value of the variable.

The role of smoothing constant $\lambda$ is very important. It is interesting by using small value of smoothing constant $\lambda$ the upper and lower control limits (UCL and LCL) make inside curvy lines from the start of lines. For example, see Fig 15, there we used $\lambda$ =0.001 and obtained inside curvy lines of UCL and LCL from the start of lines. But by using large value of smoothing constant, $\lambda$, the UCL and LCL will become straight lines from the start of lines. For example, see Fig 16, there we used $\lambda$ = 0.05 and obtained almost straight lines from the start of lines. On any EWMA CC, after fixing parameters values in control limits, it can be checked by increasing and decreasing values of $\lambda$. As more as the value of $\lambda$ near to zero, ones will get more inside curvy lines of UCL and LCL from the start of lines and as more as the value of $\lambda$ near to one, ones will get almost straight lines of UCL and LCL from the start of lines. When anyone uses $\lambda = 1$, the term $\left(\frac{\lambda}{2-\lambda}\right)\left(1 - (1 - \lambda)^{2t}\right)$ will vanish and the UCL and LCL will become completely straight lines from the start of lines, this happens because when we use $\lambda = 1$ in the EWMA statistics, $Z_{(t)} = Y_{(t)}$.

In this study, the EWMA control charts of NKP distribution based on LSE and WLSE are proposed. The R language is used to obtain the results of proposed control charts.

## 3.1. EWMA CC based on LSE of NKP distribution

In this sub-section, we used $\hat{\alpha}_{LSE(t)}$ in the place of $Y_t$ in general EWMA statistics to monitor the shape parameter based on LSE of NKP distribution. The EWMA statistics at time $t$ based on LSE of the shape parameter $\hat{\alpha}_{LSE}$ of NKP distribution is $Z_t = \lambda \hat{\alpha}_{LSE(t)} + (1 - \lambda)Z_{t-1}$, where $Z_t$ is

EWMA statistics on current time, $Z_{t-1}$ is EWMA statistics on previous time and $\lambda$ is a smoothing constant which range is $0<\lambda\leq1$. The small value of smoothing constant tells us that the less weight is given to current value of LSE of shape parameter. The large value of smoothing constant tells us that the more weight is given to previous value of LSE of shape parameter. The starting value of EWMA statistics, when $t = 1$ is the process target value, which is $Z_0 = \alpha_0$ in this study. When the process is in-control or in other words before introducing shifts in in-control process, the EWMA Control limits based on LSE are given as follows

$$UCL_{Z_t} = \alpha_0 + L\sqrt{Var(\hat{\alpha}_{LSE})\left(\frac{\lambda}{2-\lambda}\right)\left(1 - (1 - \lambda)^{2t}\right)} \tag{5}$$

$$CL_{Z_t} = \alpha_0 \tag{6}$$

$$LCL_{Z_t} = \alpha_0 - L\sqrt{Var(\hat{\alpha}_{LSE})\left(\frac{\lambda}{2-\lambda}\right)\left(1 - (1 - \lambda)^{2t}\right)} \tag{7}$$

where $L$ is the control limit multiplier. The value of $L$ controls the gap between control limits. As more as the value of $L$ uses, the gap between UCL and LCL increases and as less as the value of $L$ uses, the gap between the UCL and LCL reduces.

In this proposed chart, we monitored the shape parameter by using $\alpha_1 = \alpha_0 + \text{Shift}\sqrt{Var(\hat{\alpha}_{LSE})\left(\frac{\lambda}{2-\lambda}\right)\left(1 - (1 - \lambda)^{2t}\right)}$, where $\alpha_0$ denotes the target value of shape parameter, when the process is in-control, and $\alpha_1$ denotes a new shifted target value of shape parameter after introducing shifts in in-control process.

## 3.2. EWMA CC based on WLSE of NKP distribution

In this sub-section, we use $\hat{\alpha}_{WLSE(t)}$ in the place of $Y_t$ in general EWMA statistics to monitor the shape parameter based on WLSE of NKP distribution. The EWMA statistics at time $t$ based on WLSE of shape parameter $\hat{\alpha}_{WLSE}$ of NKP distribution is

$Z_t = \lambda\,\hat{\alpha}_{WLSE(t)} + (1 - \lambda)Z_{t-1}$ where $Z_t$ is EWMA statistics on current time and $Z_{t-1}$ is EWMA statistics on previous time. When the process is in-control or in other words before introducing shifts in in-control process, the EWMA Control limits based on WLSE are given as

$$UCL_{Z_t} = \alpha_0 + L\sqrt{Var(\hat{\alpha}_{WLSE})\left(\frac{\lambda}{2-\lambda}\right)\left(1 - (1 - \lambda)^{2t}\right)} \tag{8}$$

$$CL_{Z_t} = \alpha_0 \tag{9}$$

$$LCL_{Z_t} = \alpha_0 - L\sqrt{Var(\hat{\alpha}_{WLSE})\left(\frac{\lambda}{2-\lambda}\right)\left(1 - (1 - \lambda)^{2t}\right)} \tag{10}$$

In this proposed chart, we monitored the shape parameter by using $\alpha_1 = \alpha_0 + \text{Shift}\sqrt{Var(\hat{\alpha}_{WLSE})\left(\frac{\lambda}{2-\lambda}\right)\left(1 - (1 - \lambda)^{2t}\right)}$, where $\alpha_0$ denotes the target value of shape parameter, when the process is in-control, and $\alpha_1$ denotes a new shifted target value of shape parameter after introducing shifts in in-control process.

### 3.3. Performance of EWMA CC based on LSE and WLSE of NKP distribution

For check the performance of EWMA CC based on LSE of NKP distribution, we performed with following steps:

1. We generated random values from NKP distribution with parameters $(\alpha, \gamma, \theta) = (10,5,4)$ for $n = 40$.

2. Compute $\hat{\alpha}_{LSE}$, where $\hat{\alpha}_{LSE}$ is the LSE of shape parameter $\alpha$ of NKP distribution

3. We repeated our above two steps for 2000 times and computed $Var(\hat{\alpha}_{LSE})$.

4. We computed control limits for EWMA CC from the results that were obtained in the third step.

5. We declared the process out-of-control if $Z_t > UCL_{Z_t}$ or if $Z_t < LCL_{Z_t}$.

6. We declared the process in-control if $LCL_{Z_t} \leq Z_t \leq UCL_{Z_t}$.

7. We fixed two ARL in-control, $ARL_0$ equals 565 and 421, and we used different shifts and computed $ARL_1$ in front of shift values. Similarly, we obtained EWMA CC based on WLSE of NKP distribution.

In Table 1 and Figs 3–14, we fixed $ARL_0$ (ARL in-control) and obtained $ARL_1$ (ARL out-of-control). It observed that $ARL_1$ values were obtained by EWMA CC based on WLSE reported much better results as compared to $ARL_1$ values obtained by EWMA CC based on LSE for all shifts. We not only reported values of ARLs numerically, but we also analyzed these values through half dozen plots such as Line plot, Bar plot, Radar plot, Donut, Stacked Bar plot and Chord plot. These were plotted for comparison of EWMA CC based on LSE and EWMA CC based on WLSE.

Figs 3 and 9 display the Line plots. Both plots show that the line of ARLs of EWMA CC based on WLSE is less than the line of ARLs of EWMA CC based on LSE. This shows EWMA CC based on WLSE detected shift more quickly. The difference between the lines is shown by light yellow color.

Table 1. Comparison of EWMA CC based on LSE and WLSE with $ARL_0 = 421$ and $ARL_0 = 565$.

| Shifts | $ARL_0 = 565$ | | $ARL_0 = 421$ | |
|:---:|:---:|:---:|:---:|:---:|
| | ARL(L) | ARL(W) | ARL(L) | ARL(W) |
| 0 | 565 | 565 | 421 | 421 |
| 0.15 | 507 | 466 | 345 | 184 |
| 0.16 | 504 | 462 | 341 | 182 |
| 0.17 | 502 | 435 | 337 | 166 |
| 0.22 | 483 | 294 | 306 | 108 |
| 0.25 | 472 | 279 | 293 | 62 |
| 0.28 | 466 | 213 | 287 | 47 |
| 0.3 | 460 | 209 | 275 | 43 |
| 0.32 | 452 | 202 | 268 | 33 |
| 0.34 | 447 | 189 | 261 | 24 |
| 0.36 | 430 | 183 | 253 | 7 |
| 0.38 | 426 | 165 | 246 | 3 |
| 1.6 | 1 | 1 | 1 | 1 |

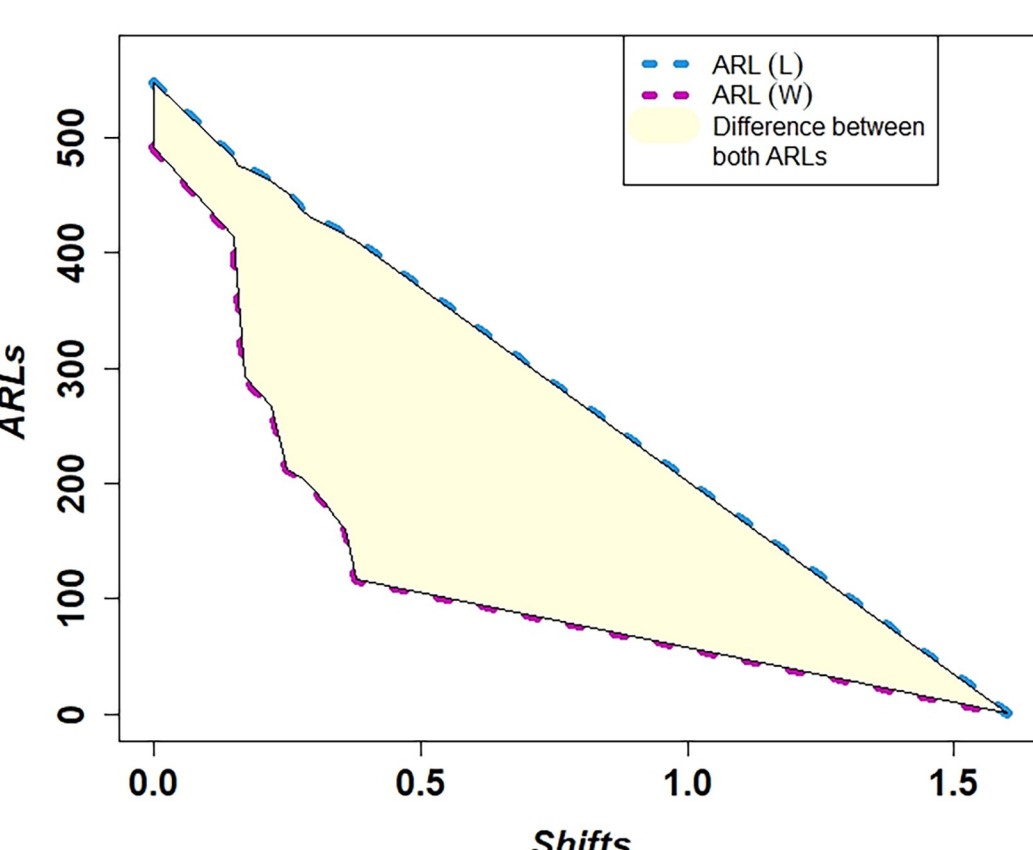

**Fig 3. Comparison of EWMA CC with $ARL_0$ = 565 by using line plot.**

Figs 4 and 10 display the bar plots. The longer bars of blue color show ARLs of EWMA CC based on LSE, whereas smaller bars of purple color show ARLs of EWMA CC based on WLSE, which shows EWMA CC based on WLSE detected shift more quickly.

Figs 5 and 11 display the radar plots. The outer longer blue line shows ARLs of EWMA CC based on LSE, whereas inner shorter purple line shows ARLs of EWMA CC based on WLSE, which shows its superiority. In front of every shift, the point of value of the ARLs of EWMA CC based on LSE and the point of value of the ARLs of EWMA CC based on WLSE is shown by blue and purple color respectively. The difference between the lines is shown by light yellow color. At first and last shift the points of both ARLs come at same place because at these shifts both ARLs are equal.

Figs 6 and 12 display the donut plots. In both plots, each shift is partitioned in two parts, such as the ARLs of EWMA CC based on LSE and the ARLs of EWMA CC based on WLSE. At shift equal to zero both ARLs are equal lengths. But when we introduced shifts, part of ARLs of EWMA CC based on LSE is larger than the part of ARLs of EWMA CC based on WLSE, which shows its supremacy of the ARLs of EWMA CC based on WLSE on the ARLs of EWMA CC based on LSE.

Figs 7 and 13 display the stacked bar plots. The longer stacked bars show ARLs of EWMA CC based on LSE, whereas the smaller the ARLs of EWMA CC based on WLSE. The sum of staked bar of the ARLs of EWMA CC based on LSE is larger than the sum of the ARLs of EWMA CC based on WLSE.

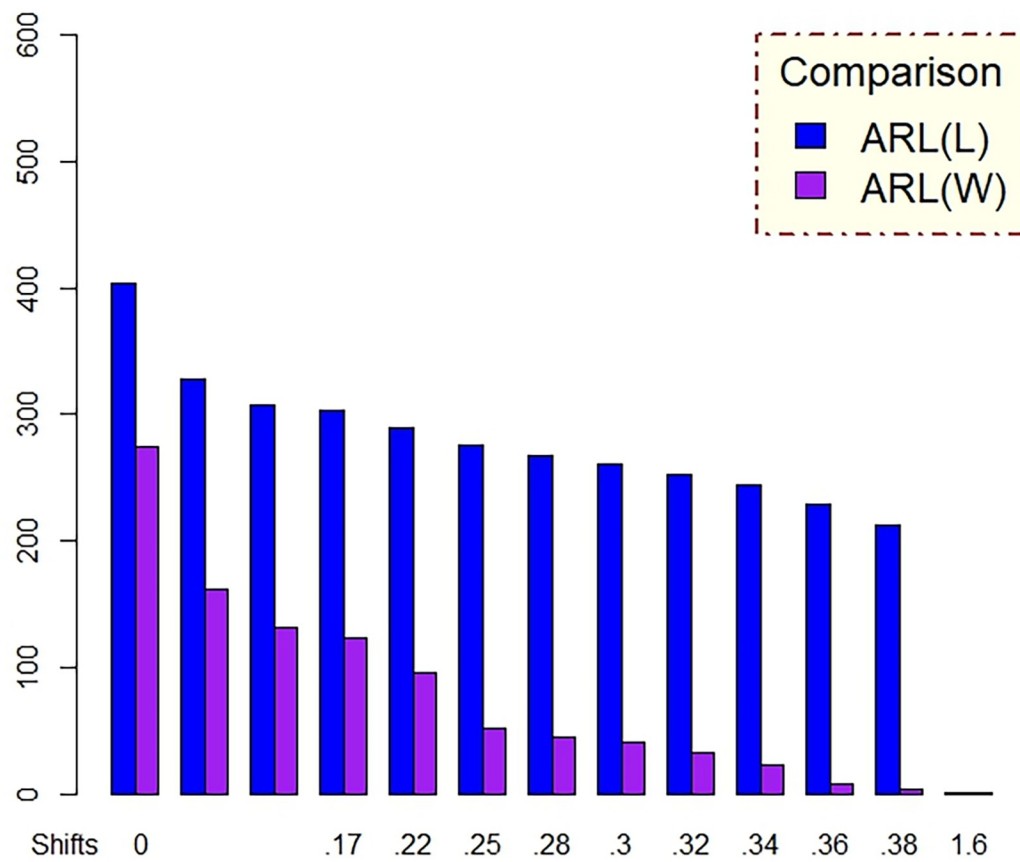

**Fig 4. Comparison of EWMA CC with $ARL_0$ = 565 by using bar plot.**

1. Figs 8 and 14 display the chord plots. The lower blue line shows the sum of ARLs of EWMA CC based on LSE and below purple line shows the sum of the ARLs of EWMA CC based on WLSE. We can see blue line is greater than purple line, which shows that the outperform of ARLs of EWMA CC based on WLSE on the ARLs of EWMA CC based on LSE. On the upper side of the plot, each shift is partitioned into two parts (lines), the line of ARLs of EWMA CC based on LSE and the line of ARLs of EWMA CC based on WLSE. The widths of all blue lines are slightly larger than the widths of purple lines, which come downward from every shift, except shifts equal to 0 and 1.6, which show the ARLs of EWMA CC based on WLSE is better than the ARLs of EWMA CC based on LSE. The key findings from the comparative analysis of EWMA control charts are summarized as follows: All the values of ARLs by using EWMA CC based on WLSE are less than EWMA CC based on LSE which is reported in Table 1 as well as from Figs 3–14.

2. The less values of EWMA CC based on WLSE shows that this control chart more quickly detect the shift as compared to other proposed control chart.

3. As is interesting result, if the values of first column are larger than the values of second column, it is necessarily the sum of first column will larger than the sum of second column. As in our case, the sum of the values of column of ARLs of EWMA CC based on LSE is larger than the sum of the values of column of ARLs of EWMA CC based on WLSE in Table 1. Fig 13 also shows the longer stacked bar is for ARLs of EWMA CC based on LSE, because its

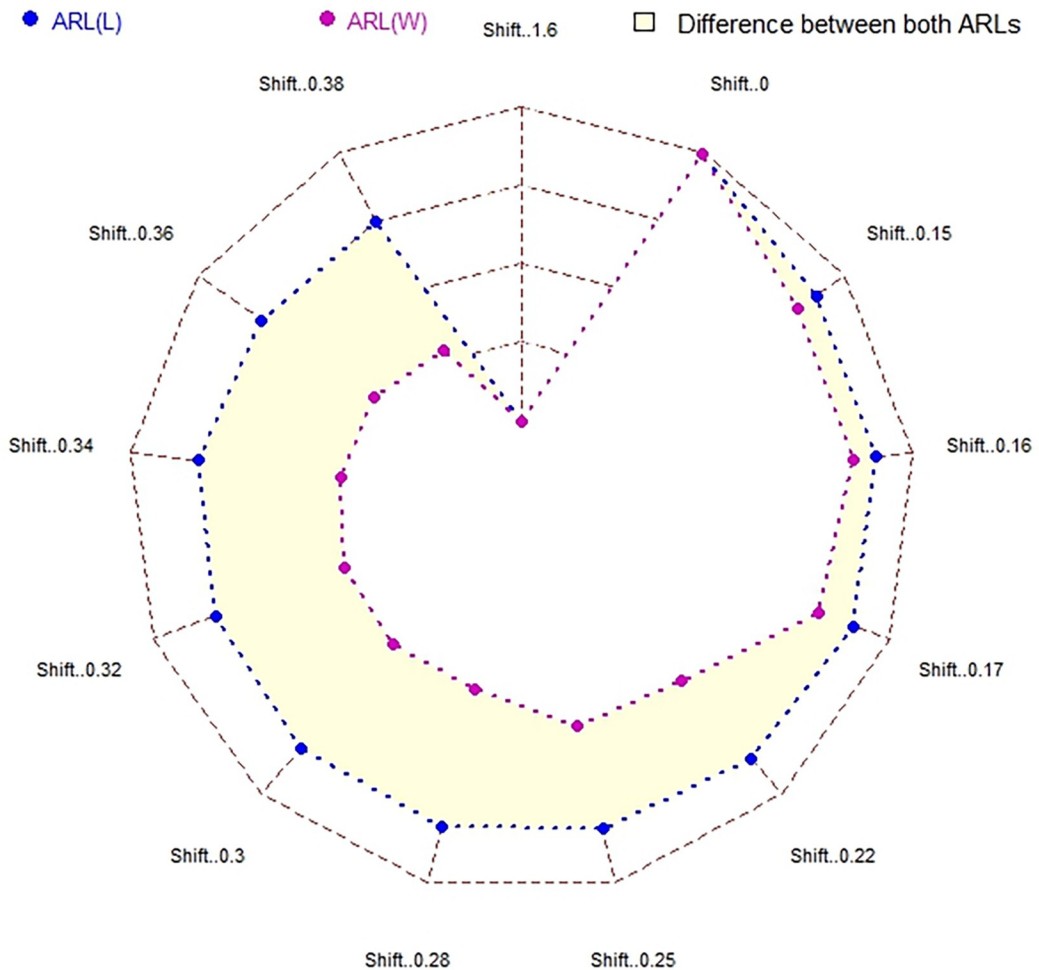

**Fig 5. Comparison of EWMA CC with $ARL_0$ = 565 by using radar plot.**

sum is large, and the smaller stacked bar is for ARLs of EWMA CC based on WLSE, because its sum is small. Additionally, Figs 8 and 14 show the lower longer blue color line is for ARLs of EWMA CC based on LSE and the lower smaller purple color line is for ARLs of EWMA CC based on WLSE.

### 3.3. Simulation study

Since the choices of parameters' values, the smoothing constant $\lambda$ and $L$ are playing important role for constructing EWMA charts. In this sub-section, we are checking how proposed charts behave depending on changes in these values and how changed the shape of the control charts.

We randomly generated observations from NKP distribution with parameter values are ($\alpha$, $\gamma$,$\theta$) = (1.5,3,2.1) to compute $\hat{\alpha}_{LSE}$. We repeated this process for 30 samples and obtained $Z_{(t)}$ and control limits (UCL and LCL) of EWMA CC. Then we plotted $Z_{(t)}$ with samples (subgroups). Same process is repeated for EWMA CC based on WLSE. The same value of $\lambda$ is used for both control charts. The results are shown in Table 2 and Fig 15. This shows that EWMA control chart based on WLSE is much better than EWMA control chart based on LSE. Because

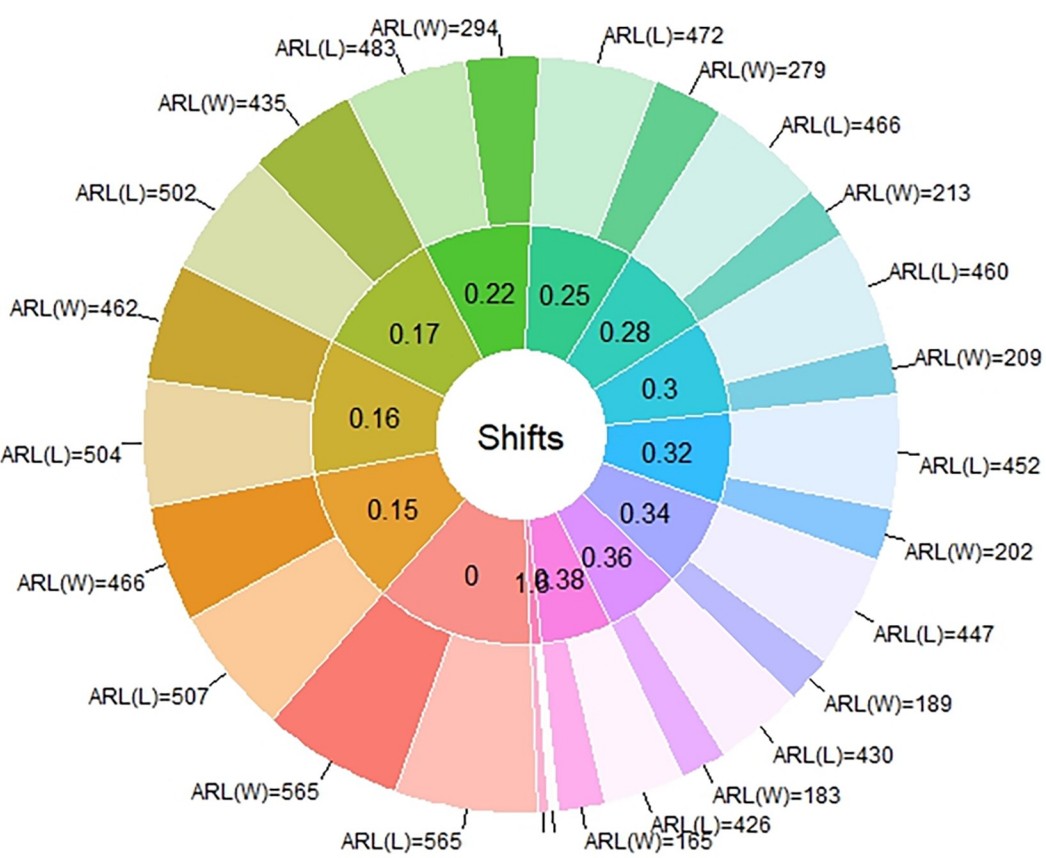

**Fig 6. Comparison of EWMA CC with $ARL_0$ = 565 by using donut plot.**

it detected last three values of EWMA statistics ($Z_{(t)}$) are out-of-control, which were in-control by using EWMA control chart based on LSE.

We can further explain the simulation results. For EWMA CC based on LSE, the first value of EWMA statistics is calculated as

$$Z_1 = \lambda \hat{\alpha}_{LSE(1)} + (1 - \lambda)Z_0 = (0.001)(2.61741) + (1 - 0.001)(1.5) = 1.50112.$$

The second value of EWMA statistics is calculated as

$$Z_2 = \lambda \hat{\alpha}_{LSE(2)} + (1 - \lambda) = Z_1(0.001)(3.83305) + (1 - 0.001)(1.50112) = 1.50345.$$

The first control limits of EWMA CC based on LSE from Eq (5) and Eq (7) are calculated as

$$UCL_{Z_1} = \alpha_0 + L\sqrt{Var(\hat{\alpha}_{LSE})\left(\frac{\lambda}{2 - \lambda}\right)\left(1 - (1 - \lambda)^2\right)}.$$

$$= 1.5 + 21\sqrt{(0.1925245)\left(\frac{0.001}{2 - 0.001}\right)\left(1 - (1 - 0.001)^2\right)} = 1.50921.$$

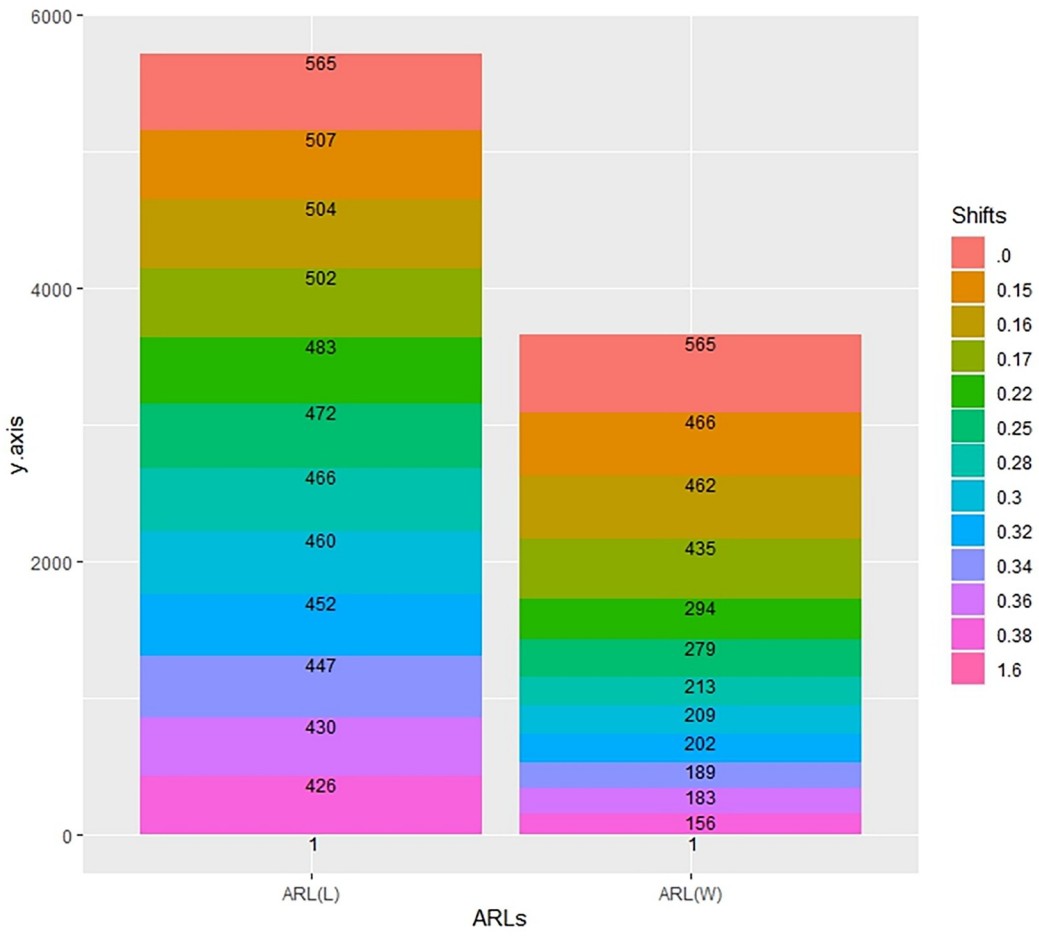

**Fig 7. Comparison of EWMA CC with $ARL_0$ = 565 by using stacked bar plot.**

$$LCL_{Z_1} = \alpha_0 - L\sqrt{Var(\hat{\alpha}_{LSE})\left(\frac{\lambda}{2-\lambda}\right)\left(1 - (1-\lambda)^2\right)}.$$

$$= 1.5 - 21\sqrt{(0.1925245)\left(\frac{0.001}{2-0.001}\right)\left(1 - (1-0.001)^2\right)} = 1.49078.$$

The second control limits of EWMA CC based on LSE from Eq (5) and Eq (7) are calculated as

$$UCL_{Z_2} = \alpha_0 + L\sqrt{Var(\hat{\alpha}_{LSE})\left(\frac{\lambda}{2-\lambda}\right)\left(1 - (1-\lambda)^4\right)}.$$

$$= 1.5 + 21\sqrt{(0.1925245)\left(\frac{0.001}{2-0.001}\right)\left(1 - (1-0.001)^4\right)} = 1.513024.$$

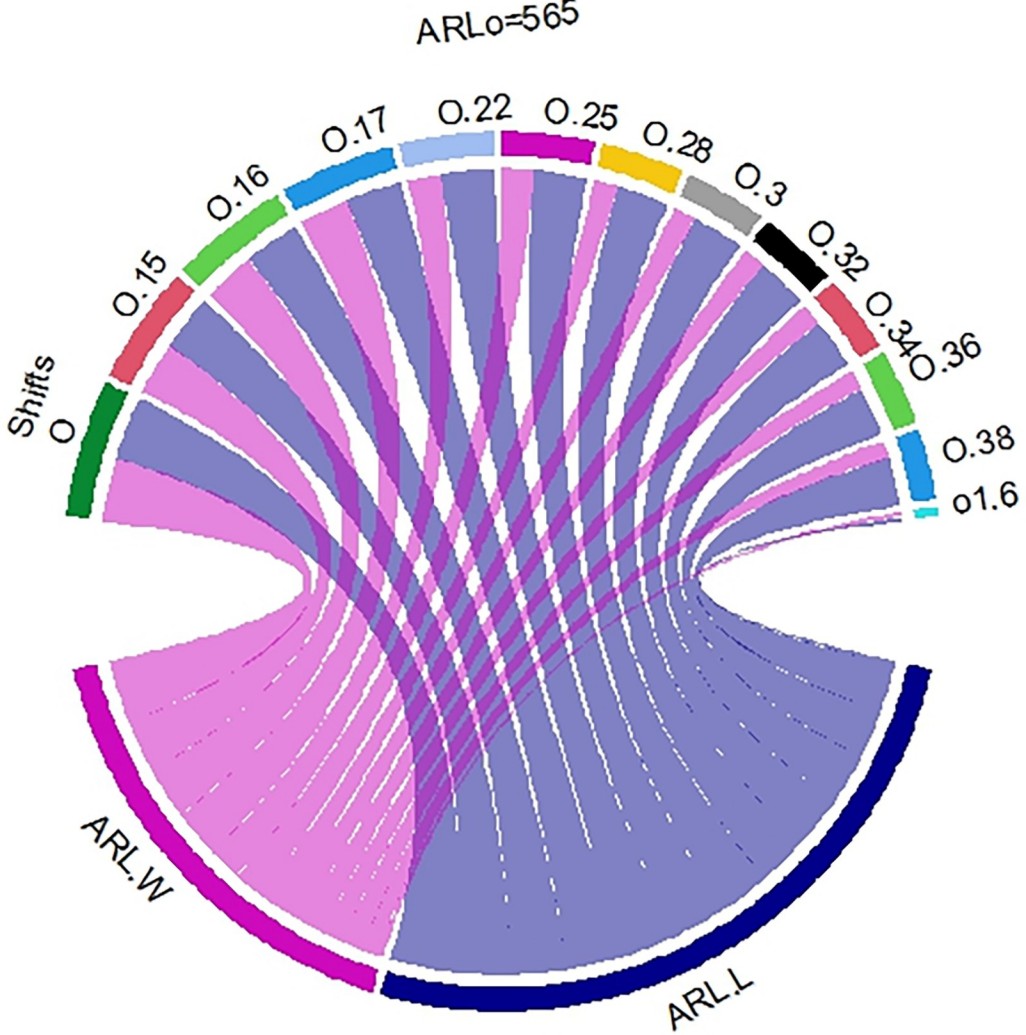

**Fig 8. Comparison of EWMA CC with $ARL_0$ = 565 by using chord plot.**

$$LCL_{Z_2} = \alpha_0 - L\sqrt{Var(\hat{\alpha}_{LSE})\left(\frac{\lambda}{2-\lambda}\right)\left(1 - (1-\lambda)^4\right)}.$$

$$= 1.5 - 21\sqrt{(0.1925245)\left(\frac{0.001}{2-0.001}\right)\left(1 - (1-0.001)^4\right)} = 1.486976.$$

For EWMA CC based on WLSE, the first value of EWMA statistics is calculated as

$$Z_1 = \lambda\hat{\alpha}_{WLSE(1)} + (1-\lambda)Z_0 = (0.001)(2.55310) + (1-0.001)(1.5) = 1.50105.$$

The second value of EWMA statistics is calculated as

$$Z_2 = \lambda\hat{\alpha}_{WLSE(2)} + (1-\lambda)Z_0 = (0.001)(3.902520) + (1-0.001)(1.50105) = 1.50345.$$

**Table 2. Simulated results of EWMA control charts based on LSE and WLSE.**

| Sample no. | EWMA Control Charts based on LSE $\lambda = 0.001$ $L = 21$ | | | | EWMA Control Charts based on WLSE $\lambda = 0.001$ $L = 14.8$ | | | |
|---|---|---|---|---|---|---|---|---|
| | $\hat{\alpha}_{LSE(t)}$ | $Z_t$ | $UCL_{Z_t}$ | $LCL_{Z_t}$ | $\hat{\alpha}_{WLSE(t)}$ | $Z_t$ | $UCL_{Z_t}$ | $LCL_{Z_t}$ |
| 1 | 2.61741 | 1.50112 | 1.509214 | 1.490786 | 2.55310 | 1.50105 | 1.507500 | 1.492500 |
| 2 | 3.83305 | 1.50345 | 1.513024 | 1.486976 | 3.90252 | 1.50345 | 1.510601 | 1.489399 |
| 3 | 3.46227 | 1.50541 | 1.515944 | 1.484056 | 2.63681 | 1.50459 | 1.512977 | 1.487023 |
| 4 | 2.90389 | 1.50681 | 1.518401 | 1.481599 | 3.26577 | 1.50635 | 1.514978 | 1.485022 |
| 5 | 2.71957 | 1.50802 | 1.520563 | 1.479437 | 2.54560 | 1.50739 | 1.516737 | 1.483263 |
| 6 | 3.42504 | 1.50994 | 1.522514 | 1.477486 | 2.82214 | 1.50870 | 1.518325 | 1.481675 |
| 7 | 2.57402 | 1.51100 | 1.524306 | 1.475694 | 3.38983 | 1.51058 | 1.519784 | 1.480216 |
| 8 | 2.94147 | 1.51243 | 1.525971 | 1.474029 | 2.78104 | 1.51185 | 1.521139 | 1.478861 |
| 9 | 2.63721 | 1.51356 | 1.527533 | 1.472467 | 2.54658 | 1.51289 | 1.522410 | 1.477590 |
| 10 | 3.19419 | 1.51524 | 1.529008 | 1.470992 | 3.92129 | 1.51530 | 1.523611 | 1.476389 |
| 11 | 3.67385 | 1.51740 | 1.530408 | 1.469592 | 2.66763 | 1.51645 | 1.524751 | 1.475249 |
| 12 | 2.70403 | 1.51858 | 1.531744 | 1.468256 | 2.72616 | 1.51766 | 1.525839 | 1.474161 |
| 13 | 2.89505 | 1.51996 | 1.533024 | 1.466976 | 2.75635 | 1.51890 | 1.526880 | 1.473120 |
| 14 | 3.91710 | 1.52236 | 1.534254 | 1.465746 | 2.53916 | 1.51992 | 1.527881 | 1.472119 |
| 15 | 3.46633 | 1.52430 | 1.535438 | 1.464562 | 2.51499 | 1.52091 | 1.528845 | 1.471155 |
| 16 | 2.64823 | 1.52542 | 1.536582 | 1.463418 | 2.48926 | 1.52188 | 1.529777 | 1.470223 |
| 17 | 3.71600 | 1.52761 | 1.53769 | 1.46231 | 2.48397 | 1.52284 | 1.530678 | 1.469322 |
| 18 | 2.67217 | 1.52876 | 1.538763 | 1.461237 | 3.16810 | 1.52449 | 1.531551 | 1.468449 |
| 19 | 2.74466 | 1.52997 | 1.539805 | 1.460195 | 2.51566 | 1.52548 | 1.532400 | 1.467600 |
| 20 | 2.52190 | 1.53097 | 1.540819 | 1.459181 | 2.47562 | 1.52643 | 1.533225 | 1.466775 |
| 21 | 2.90834 | 1.53234 | 1.541806 | 1.458194 | 3.90604 | 1.52881 | 1.534029 | 1.465971 |
| 22 | 2.82703 | 1.53364 | 1.542769 | 1.457231 | 2.93137 | 1.53021 | 1.534812 | 1.465188 |
| 23 | 2.99892 | 1.53510 | 1.543708 | 1.456292 | 3.99452 | 1.53268 | 1.535577 | 1.464423 |
| 24 | 2.83795 | 1.53641 | 1.544626 | 1.455374 | 3.73056 | 1.53488 | 1.536324 | 1.463676 |
| 25 | 2.47856 | 1.53735 | 1.545524 | 1.454476 | 2.36164 | 1.53570 | 1.537055 | 1.462945 |
| 26 | 3.24075 | 1.53905 | 1.546403 | 1.453597 | 2.68603 | 1.53685 | 1.537770 | 1.462230 |
| 27 | 3.89867 | 1.54141 | 1.547263 | 1.452737 | 2.79389 | 1.53811 | 1.538470 | 1.461530 |
| 28 | 3.19070 | 1.54306 | 1.548106 | 1.451894 | 3.37808 | 1.53995 | 1.539157 | 1.460843 |
| 29 | 2.75618 | 1.54427 | 1.548934 | 1.451066 | 3.08305 | 1.54149 | 1.539830 | 1.460170 |
| 30 | 3.21773 | 1.54595 | 1.549746 | 1.450254 | 2.88009 | 1.54283 | 1.540491 | 1.459509 |

The first control limits of EWMA CC based on WLSE from Eq (8) and Eq (10) are calculated as

$$UCL_{Z_1} = \alpha_0 + L\sqrt{Var(\hat{\alpha}_{WLSE})\left(\frac{\lambda}{2-\lambda}\right)\left(1 - (1-\lambda)^2\right)}.$$

$$= 1.5 + 14.8\sqrt{(0.2568061)\left(\frac{0.001}{2-0.001}\right)\left(1 - (1-0.001)^2\right)} = 1.50750.$$

$$LCL_{Z_1} = \alpha_0 - L\sqrt{Var(\hat{\alpha}_{WLSE})\left(\frac{\lambda}{2-\lambda}\right)\left(1 - (1-\lambda)^2\right)}.$$

**Table 3. Simulated results of EWMA control charts using LSE and WLSE for data set 1.**

| Serial no. | Sample no. | EWMA Control Charts using LSE $\lambda = 0.5\ L = 14$ | | | | EWMA Control Charts using WLSE $\lambda = 0.5\ L = 10$ | | | |
|---|---|---|---|---|---|---|---|---|---|
| | | $\hat{\alpha}_{LSE(t)}$ | $Z_t$ | $UCL_{Z_t}$ | $LCL_{Z_t}$ | $\hat{\alpha}_{WLSE(t)}$ | $Z_t$ | $UCL_{Z_t}$ | $LCL_{Z_t}$ |
| 1 | 1 | 132.0598821 | 146.0299 | 187.4379 | 132.5621 | 132.0584356 | 146.0292 | 182.7674 | 137.2326 |
| 2 | 2 | 128.9799126 | 137.5049 | 190.6765 | 129.3235 | 128.9532298 | 137.4912 | 185.4547 | 134.5453 |
| 3 | 3 | 139.9999998 | 138.7525 | 191.4341 | 128.5659 | 139.9999973 | 138.7456 | 186.0833 | 133.9167 |
| 4 | 4 | 139.9999960 | 139.3762 | 191.6206 | 128.3794 | 139.9999998 | 139.3728 | 186.2381 | 133.7619 |
| 5 | 5 | 139.9999794 | 139.6881 | 191.6671 | 128.3329 | 129.9893287 | 134.6811 | 186.2767 | 133.7233 |
| 6 | 6 | 139.9999978 | 139.8441 | 191.6787 | 128.3213 | 139.9999741 | 137.3405 | 186.2863 | 133.7137 |
| 7 | 7 | 141.3907158 | 140.6174 | 191.6816 | 128.3184 | 141.3908186 | 139.3657 | 186.2887 | 133.7113 |
| 8 | 8 | 136.2045678 | 138.4110 | 191.6823 | 128.3177 | 136.2049344 | 137.7853 | 186.2893 | 133.7107 |
| 9 | 9 | 136.2045678 | 137.3078 | 191.6825 | 128.3175 | 136.2049344 | 136.9951 | 186.2895 | 133.7105 |
| 10 | 10 | 139.9999999 | 138.6539 | 191.6825 | 128.3175 | 139.9999992 | 138.4976 | 186.2895 | 133.7105 |
| 11 | 11 | 139.9999998 | 139.3269 | 191.6826 | 128.3174 | 139.9999999 | 139.2488 | 186.2895 | 133.7105 |
| 12 | 12 | 139.9999794 | 139.6635 | 191.6826 | 128.3174 | 129.9893287 | 134.6191 | 186.2895 | 133.7105 |
| 13 | 13 | 128.9799126 | 134.3217 | 191.6826 | 128.3174 | 128.9532298 | 131.7861 | 186.2895 | 133.7105 |
| 14 | 14 | 139.9999999 | 137.1608 | 191.6826 | 128.3174 | 139.9999996 | 135.8931 | 186.2895 | 133.7105 |
| 15 | 15 | 142.4295232 | 139.7952 | 191.6826 | 128.3174 | 142.4296193 | 139.1614 | 186.2895 | 133.7105 |
| 16 | 16 | 138.2767724 | 139.0360 | 191.6826 | 128.3174 | 138.2767565 | 138.7191 | 186.2895 | 133.7105 |
| 17 | 17 | 134.1293628 | 136.5827 | 191.6826 | 128.3174 | 134.1300202 | 136.4245 | 186.2895 | 133.7105 |
| 18 | 18 | 133.0997337 | 134.8412 | 191.6826 | 128.3174 | 133.1021466 | 134.7633 | 186.2895 | 133.7105 |
| 19 | 19 | 139.9999794 | 137.4206 | 191.6826 | 128.3174 | 129.9893287 | 132.3763 | 186.2895 | 133.7105 |
| 20 | 20 | 139.9999999 | 138.7103 | 191.6826 | 128.3174 | 139.9999998 | 136.1882 | 186.2895 | 133.7105 |
| 21 | 21 | 139.9999998 | 139.3552 | 191.6826 | 128.3174 | 139.9999973 | 138.0941 | 186.2895 | 133.7105 |
| 22 | 22 | 138.2767724 | 138.8160 | 191.6826 | 128.3174 | 138.2767565 | 138.1854 | 186.2895 | 133.7105 |
| 23 | 23 | 128.9799126 | 133.8979 | 191.6826 | 128.3174 | 128.9532298 | 133.5693 | 186.2895 | 133.7105 |
| 24 | 24 | 139.9999794 | 136.9490 | 191.6826 | 128.3174 | 129.9893287 | 131.7793 | 186.2895 | 133.7105 |
| 25 | 25 | 139.9999999 | 138.4745 | 191.6826 | 128.3174 | 139.9999998 | 135.8897 | 186.2895 | 133.7105 |
| 26 | 26 | 142.4295232 | 140.4520 | 191.6826 | 128.3174 | 142.4296193 | 139.1596 | 186.2895 | 133.7105 |
| 27 | 27 | 133.0997337 | 136.7759 | 191.6826 | 128.3174 | 133.1021466 | 136.1309 | 186.2895 | 133.7105 |
| 28 | 28 | 139.9999999 | 138.3879 | 191.6826 | 128.3174 | 139.9999997 | 138.0655 | 186.2895 | 133.7105 |
| 29 | 29 | 139.9999999 | 139.1940 | 191.6826 | 128.3174 | 139.9999998 | 139.0327 | 186.2895 | 133.7105 |
| 30 | 30 | 128.9799126 | 134.0869 | 191.6826 | 128.3174 | 128.9532298 | 133.9930 | 186.2895 | 133.7105 |
| 31 | 31 | 134.1293628 | 134.1082 | 191.6826 | 128.3174 | 134.1300202 | 134.0615 | 186.2895 | 133.7105 |
| 32 | 32 | 136.2045678 | 135.1564 | 191.6826 | 128.3174 | 136.2049344 | 135.1332 | 186.2895 | 133.7105 |
| 33 | 33 | 139.9999999 | 137.5782 | 191.6826 | 128.3174 | 139.9999998 | 137.5666 | 186.2895 | 133.7105 |
| 34 | 34 | 139.9999999 | 138.7891 | 191.6826 | 128.3174 | 139.9999996 | 138.7833 | 186.2895 | 133.7105 |
| 35 | 35 | 135.1661177 | 136.9776 | 191.6826 | 128.3174 | 135.1658953 | 136.9746 | 186.2895 | 133.7105 |
| 36 | 36 | 134.1293628 | 135.5535 | 191.6826 | 128.3174 | 134.1300202 | 135.5523 | 186.2895 | 133.7105 |
| 37 | 37 | 139.9999794 | 137.7767 | 191.6826 | 128.3174 | 129.9893287 | 132.7708 | 186.2895 | 133.7105 |
| 38 | 38 | 139.9999978 | 138.8884 | 191.6826 | 128.3174 | 139.9999741 | 136.3854 | 186.2895 | 133.7105 |
| 39 | 39 | 139.9999993 | 139.4442 | 191.6826 | 128.3174 | 139.9999917 | 138.1927 | 186.2895 | 133.7105 |
| 40 | 40 | 139.9999372 | 139.7221 | 191.6826 | 128.3174 | 131.0976480 | 134.6452 | 186.2895 | 133.7105 |

**Fig 9. Comparison of EWMA CC with $ARL_0$ = 421 by using line plot.**

**Table 4. Simulated results of EWMA control charts using LSE and WLSE for data set 2.**

| Serial no. | Sample no. | EWMA Control Charts using LSE $\lambda = 0.23$ $L = 4.7$ | | | | EWMA Control Charts using WLSE $\lambda = 0.23$ $L = 3.9$ | | | |
|---|---|---|---|---|---|---|---|---|---|
| | | $\hat{\alpha}_{LSE(t)}$ | $Z_t$ | $UCL_{Z_t}$ | $LCL_{Z_t}$ | $\hat{\alpha}_{WLSE(t)}$ | $Z_t$ | $UCL_{Z_t}$ | $LCL_{Z_t}$ |
| 1 | 1 | 158.9999 | 156.6900 | 157.8239 | 154.1761 | 159.0000 | 156.6900 | 157.5134 | 154.4866 |
| 2 | 2 | 159.0000 | 157.2213 | 158.3020 | 153.6980 | 159.0001 | 157.2214 | 157.9101 | 154.0899 |
| 3 | 3 | 159.0001 | 157.6304 | 158.5434 | 153.4566 | 159.0001 | 157.6305 | 158.1104 | 153.8896 |
| 4 | 4 | 158.9999 | 157.9454 | 158.6762 | 153.3238 | 158.9999 | 157.9454 | 158.2206 | 153.7794 |
| 5 | 5 | 159.0001 | 158.1880 | 158.7519 | 153.2481 | 159.0000 | 158.1880 | 158.2835 | 153.7165 |
| 6 | 6 | 159.0001 | 158.3747 | 158.7959 | 153.2041 | 159.0001 | 158.3748 | 158.3199 | 153.6801 |
| 7 | 7 | 155.2080 | 157.6464 | 158.8216 | 153.1784 | 155.2083 | 157.6465 | 158.3413 | 153.6587 |
| 8 | 8 | 161.4060 | 158.5111 | 158.8367 | 153.1633 | 161.4062 | 158.5112 | 158.3538 | 153.6462 |

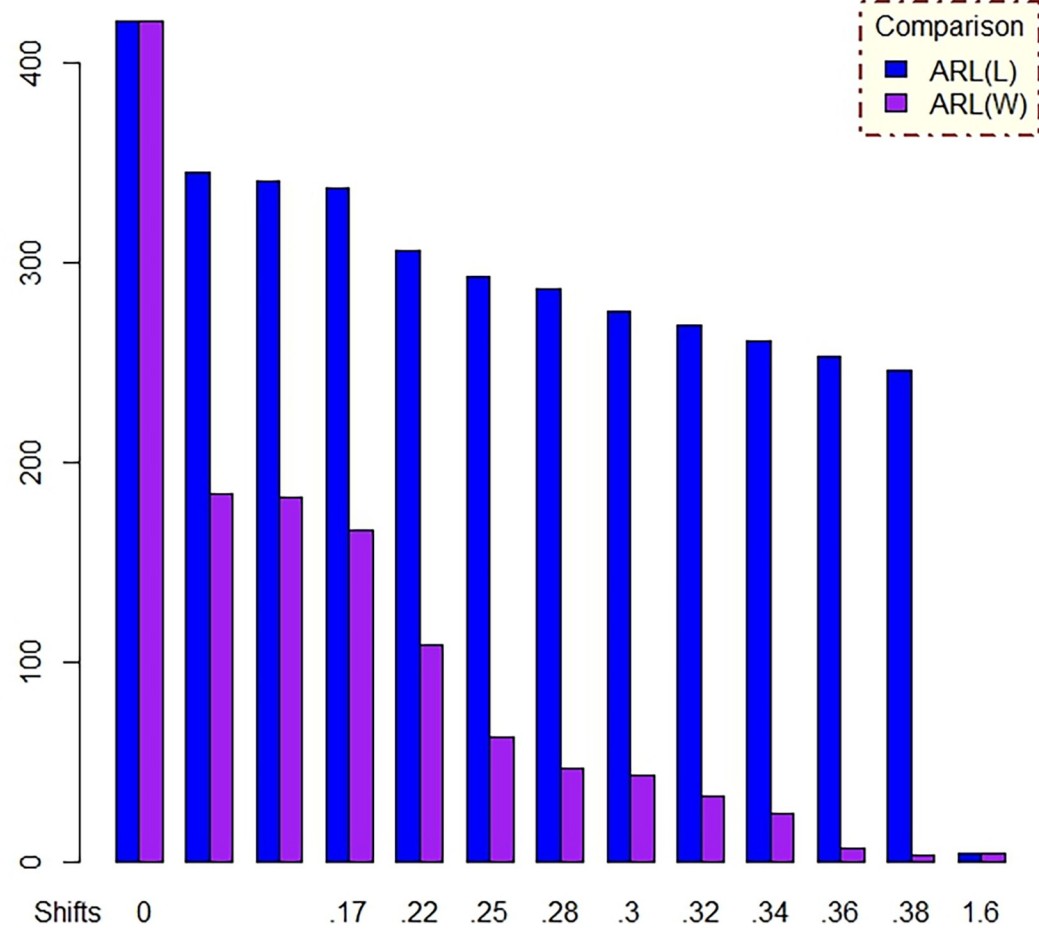

**Fig 10. Comparison of EWMA CC with $ARL_0$ = 421 by using bar plot.**

$$= 1.5 - 14.8\sqrt{(0.2568061)\left(\frac{0.001}{2 - 0.001}\right)\left(1 - (1 - 0.001)^2\right)} = 1.49250.$$

The second control limits of EWMA CC based on WLSE from Eq (8) and Eq (10) are calculated as

$$UCL_{Z_2} = \alpha_0 + L\sqrt{Var(\hat{\alpha}_{WLSE})\left(\frac{\lambda}{2 - \lambda}\right)\left(1 - (1 - \lambda)^4\right)}.$$

$$= 1.5 + 14.8\sqrt{(0.2568061)\left(\frac{0.001}{2 - 0.001}\right)\left(1 - (1 - 0.001)^4\right)} = 1.510601.$$

$$LCL_{Z_2} = \alpha_0 - L\sqrt{Var(\hat{\alpha}_{WLSE})\left(\frac{\lambda}{2 - \lambda}\right)\left(1 - (1 - \lambda)^4\right)}.$$

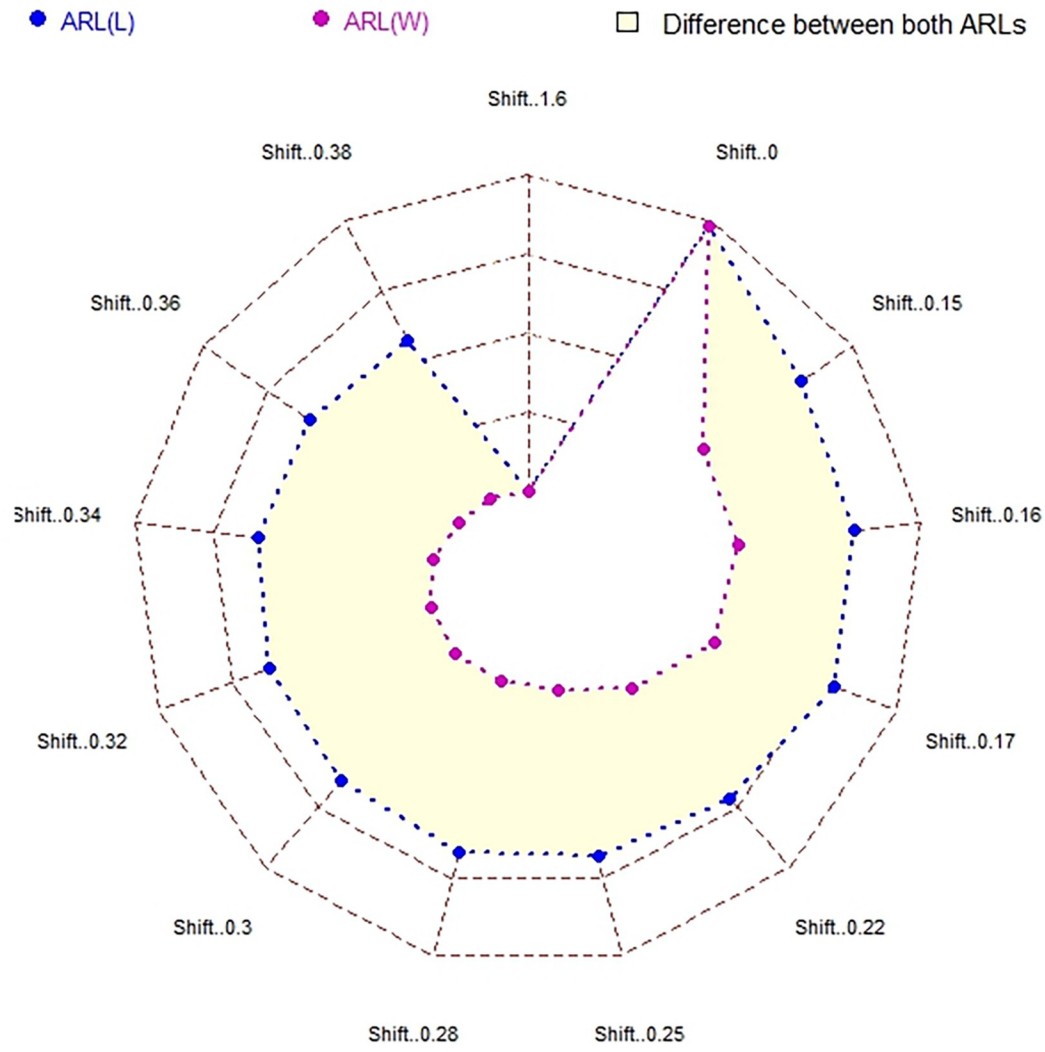

**Fig 11. Comparison of EWMA CC with $ARL_0$ = 421 by using radar plot.**

$$= 1.5 - 14.8\sqrt{(0.2568061)\left(\frac{0.001}{2 - 0.001}\right)\left(1 - (1 - 0.001)^4\right)} = 1.489399.$$

and so on.

It is notable if we ignore the term $\left(\frac{\lambda}{2-\lambda}\right)\left(1 - (1 - \lambda)^{2t}\right)$ from the control limits of both proposed control charts, the remaining control limits will stable for all values of $t$, because there is no term left there having $t$, as

$$UCL_{Z_t} = \alpha_0 + L\sqrt{Var(\hat{\alpha}_{LSE})\left(\frac{\lambda}{2 - \lambda}\right)}.$$

$$LCL_{Z_t} = \alpha_0 - L\sqrt{Var(\hat{\alpha}_{LSE})\left(\frac{\lambda}{2 - \lambda}\right)}.$$

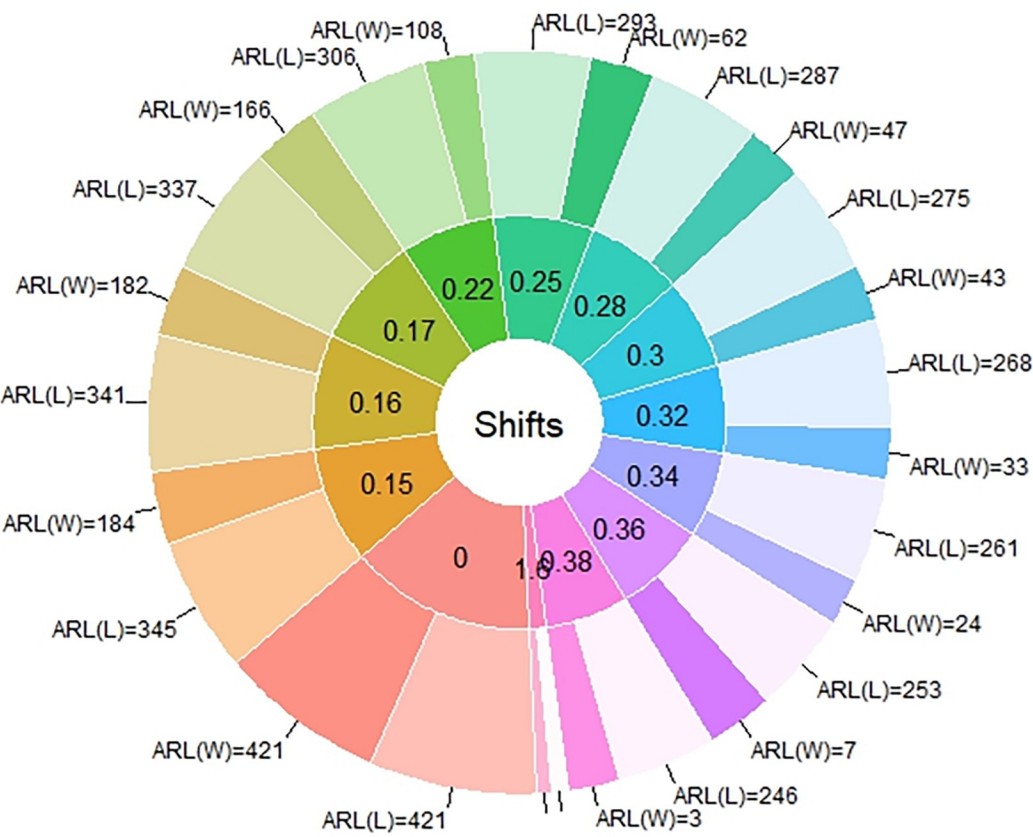

**Fig 12. Comparison of EWMA CC with $ARL_0$ = 421 by using donut plot.**

$$UCL_{Z_t} = \alpha_0 + L\sqrt{Var(\hat{\alpha}_{WLSE})\left(\frac{\lambda}{2-\lambda}\right)}.$$

$$LCL_{Z_t} = \alpha_0 - L\sqrt{Var(\hat{\alpha}_{WLSE})\left(\frac{\lambda}{2-\lambda}\right)}.$$

## 1.4 Limitations of the proposed control charts

This sub-section reported some limitations that should be kept in mind before using these control charts.

1. The values of parameter $\alpha$ should be greater than the highest value of data, always, which uses for creating proposed control charts. The reason behind to this philosophy is that NKP distribution has domain/support is $0 < y < \alpha$.

2. The smoothing constant $\lambda$ should lies between 0 and 1 in proposed control charts as it is an essential limitation of general EWMA control chart.

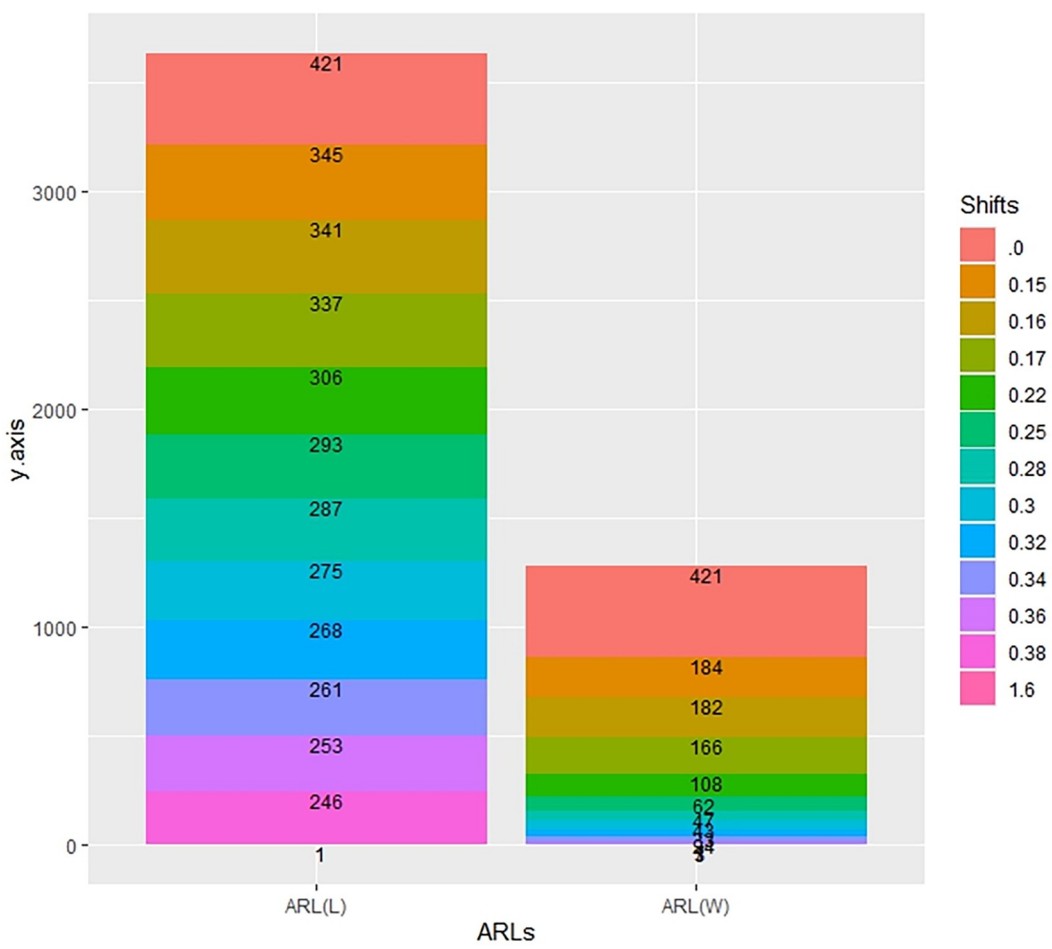

**Fig 13. Comparison of EWMA CC with $ARL_0$ = 421 by using stacked bar plot.**

3. For any data, the EWMA CC based on WLSE is always better than EWMA CC based on LSE. There are two philosophies which run behind this argument. First philosophy is that WLSE is weighted form (modified form) of least square estimator. The second philosophy is that we have seen in its baseline on [1] that all estimators with all estimation methods, the WLSE is better than LSE.

## 4. Application

In this section, the performance of the proposed control charts is shown based on two real data sets.

**Data set 1:** The first data set related to "Electronic Component Failure Time". This data is from Table 3 E.1 of [26]. Table 3 proposes the simulated results of EWMA Control Charts using LSE and WLSE for this data set. Some details relevant to calculations adopted for this data set are follows:

For EWMA CC based on LSE, the first value of EWMA statistics is calculated as

$$Z_1 = \lambda\,\hat{\alpha}_{LSE(1)} + (1 - \lambda)Z_0 = (0.5)(132.0598) + (1 - 0.5)(160.00001) = 146.0299.$$

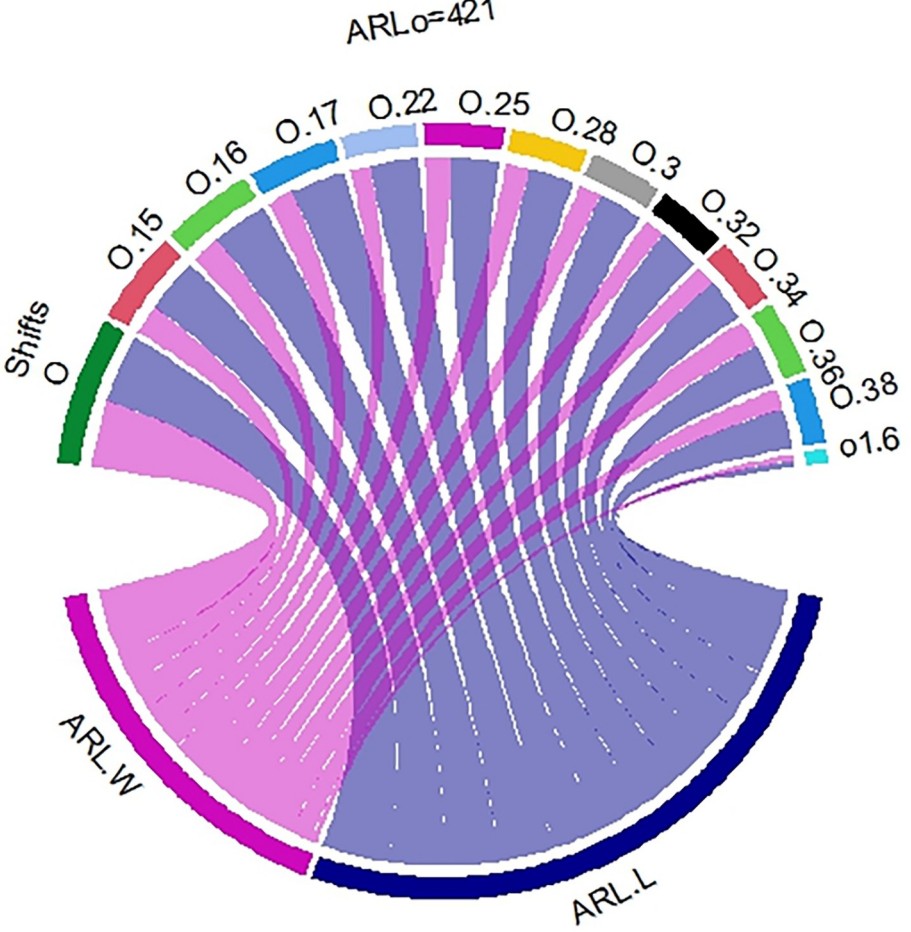

**Fig 14. Comparison of EWMA CC with $ARL_0$ = 421 by using chord plot.**

The second value of EWMA statistics is calculated as

$$Z_2 = \lambda\, \hat{\alpha}_{LSE(2)} + (1 - \lambda)Z_1 = (0.5)(128.9799) + (1 - 0.5)Z_1 = 137.5049.$$

The first control limits of EWMA CC based on LSE from Eq (5) and Eq (7) are calculated as

$$UCL_{Z_1} = \alpha_0 + L\sqrt{Var(\hat{\alpha}_{LSE})\left(\frac{\lambda}{2 - \lambda}\right)\left(1 - (1 - \lambda)^2\right)}.$$

$$= 160.00001 + 14\sqrt{(15.36405)\left(\frac{0.5}{2 - 0.5}\right)\left(1 - (1 - 0.5)^2\right)} = 187.4379.$$

$$LCL_{Z_1} = \alpha_0 - L\sqrt{Var(\hat{\alpha}_{LSE})\left(\frac{\lambda}{2 - \lambda}\right)\left(1 - (1 - \lambda)^2\right)}.$$

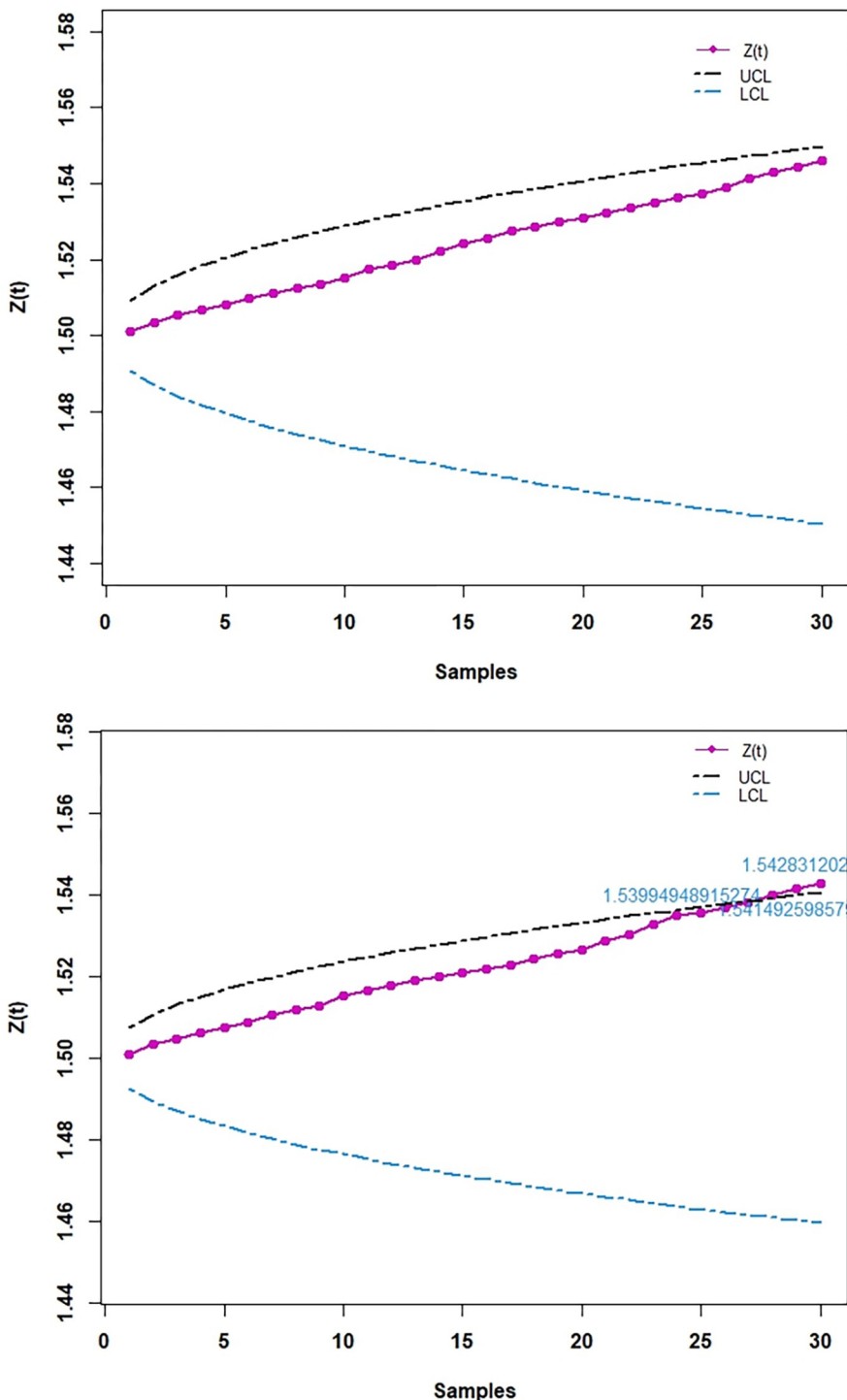

**Fig 15. EWMA CC simulated.** (A) Based on LSE. (B) Based on WLSE.

$$= 160.00001 - 14\sqrt{(15.36405)\left(\frac{0.5}{2 - 0.5}\right)\left(1 - (1 - 0.5)^2\right)} = 132.5621$$

The second control limits of EWMA CC based on LSE from Eq (5) and Eq (7) are calculated as

$$UCL_{Z_2} = \alpha_0 + L\sqrt{Var(\hat{\alpha}_{LSE})\left(\frac{\lambda}{2 - \lambda}\right)\left(1 - (1 - \lambda)^4\right)}.$$

$$= 160.00001 + 14\sqrt{(15.36405)\left(\frac{0.5}{2 - 0.5}\right)\left(1 - (1 - 0.5)^4\right)} = 190.6765.$$

$$LCL_{Z_2} = \alpha_0 - L\sqrt{Var(\hat{\alpha}_{LSE})\left(\frac{\lambda}{2 - \lambda}\right)\left(1 - (1 - \lambda)^4\right)}.$$

$$LCL_{Z_2} = 160.00001 - 14\sqrt{(15.36405)\left(\frac{0.5}{2 - 0.5}\right)\left(1 - (1 - 0.5)^4\right)} = 129.3235.$$

For EWMA CC based on WLSE, the first value of EWMA statistics is calculated as

$$Z_1 = \lambda\,\hat{\alpha}_{WLSE(1)} + (1 - \lambda)Z_0.$$

$$= (0.5)(132.0584) + (1 - 0.5)(160.00001) = 146.0292.$$

The second value of EWMA statistics is calculated as

$$Z_2 = \lambda\,\hat{\alpha}_{WLSE(2)} + (1 - \lambda)Z_1.$$

$$= (0.5)(128.9532) + (1 - 0.5)(146.0292) = 137.4912.$$

The first control limits of EWMA CC based on WLSE from Eq (8) and Eq (10) is calculated as

$$UCL_{Z_1} = \alpha_0 + L\sqrt{Var(\hat{\alpha}_{WLSE})\left(\frac{\lambda}{2 - \lambda}\right)\left(1 - (1 - \lambda)^2\right)}.$$

$$= 160.00001 + 10\sqrt{(20.73419)\left(\frac{0.5}{2 - 0.5}\right)\left(1 - (1 - 0.5)^2\right)} = 182.7674.$$

$$LCL_{Z_1} = \alpha_0 - L\sqrt{Var(\hat{\alpha}_{WLSE})\left(\frac{\lambda}{2 - \lambda}\right)\left(1 - (1 - \lambda)^2\right)}.$$

$$= 160.00001 - 10\sqrt{(20.73419)\left(\frac{0.5}{2-0.5}\right)\left(1-(1-0.5)^2\right)} = 137.2326.$$

The second control limits of EWMA CC based on WLSE from Eq (8) and Eq (10) are calculated as

$$UCL_{Z_2} = \alpha_0 + L\sqrt{Var(\hat{\alpha}_{WLSE})\left(\frac{\lambda}{2-\lambda}\right)\left(1-(1-\lambda)^4\right)}.$$

$$= 160.00001 - 10\sqrt{(20.73419)\left(\frac{0.5}{2-0.5}\right)\left(1-(1-0.5)^4\right)} = 185.4547.$$

$$LCL_{Z_2} = \alpha_0 - L\sqrt{Var(\hat{\alpha}_{WLSE})\left(\frac{\lambda}{2-\lambda}\right)\left(1-(1-\lambda)^4\right)}.$$

$$= 160.00001 - 10\sqrt{(20.73419)\left(\frac{0.5}{2-0.5}\right)\left(1-(1-0.5)^4\right)} = 134.5453.$$

**Data set 2:** The second data set related to "Yield strengths of circular tubes with end caps". The data set 2 is from Exercise 3.5 of [26]. Table 4 proposes the simulated results of EWMA Control Charts using LSE and WLSE for this data set.

In Tables 3 and 4, and Figs 16–19 the values obtained of control limits (UCL and LCL) of both EWMA control charts and plotted respective EWMA statistics, $Z_t$, corresponding to subgroups (sample) for both real data sets. For the first real data set, EWMA control chart based on WLSE detected five EWMA statistics. For the second real data set, EWMA control chart based on WLSE detected two EWMA statistics out-of-control which were in-control by using EWMA control charts based on LSE.

Some details relevant to calculations adopted for data set 2 are follows:

For EWMA CC based on LSE, the first value of EWMA statistics is calculated as

$$Z_1 = \lambda \, \hat{\alpha}_{LSE(1)} + (1-\lambda)Z_0.$$

$$= (0.23)(158.9999) + (1-0.23)(156.00001) = 156.6900.$$

The second value of EWMA statistics is calculated as

$$Z_2 = \lambda \, \hat{\alpha}_{LSE(2)} + (1-\lambda)Z_1.$$

$$= (0.23)(159.0000) + (1-0.23)(156.6900) = 157.2213.$$

The first control limits of EWMA CC based on LSE from Eq (5) and Eq (7) are calculated as

$$UCL_{Z_1} = \alpha_0 + L\sqrt{Var(\hat{\alpha}_{LSE})\left(\frac{\lambda}{2-\lambda}\right)\left(1-(1-\lambda)^2\right)}.$$

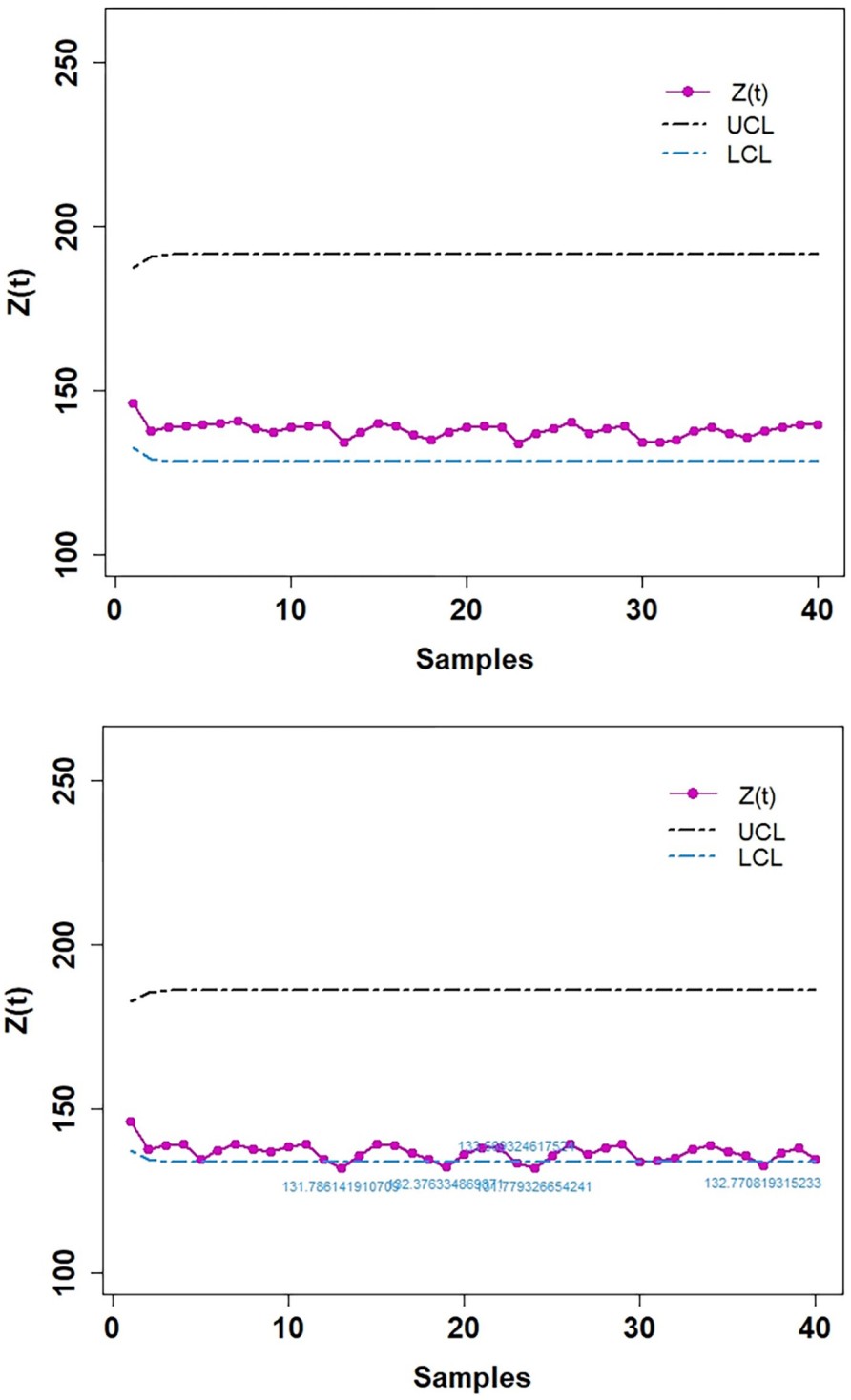

**Fig 16. EWMA CC for data set 1.** (a) Based on LSE. (b) Based on WLSE.

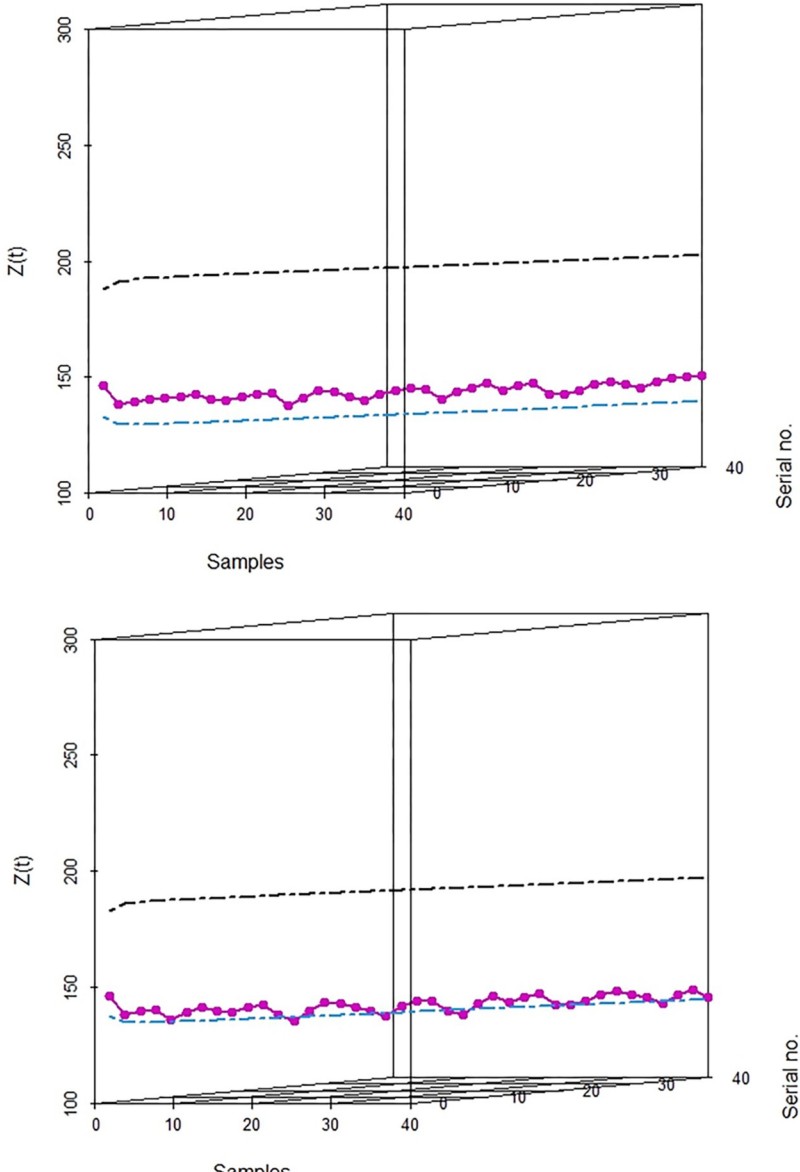

**Fig 17. 3D plot EWMA CC for data set 1.** (a) Based on LSE. (b) Based on WLSE.

$$= 156.00001 + 4.7\sqrt{(2.846888)\left(\frac{0.23}{2-0.23}\right)\left(1-(1-0.23)^2\right)} = 157.8240.$$

$$LCL_{Z_2} = \alpha_0 - L\sqrt{Var(\hat{\alpha}_{LSE})\left(\frac{\lambda}{2-\lambda}\right)\left(1-(1-\lambda)^2\right)}.$$

$$= 156.00001 - 4.7\sqrt{(2.846888)\left(\frac{0.23}{2-0.23}\right)\left(1-(1-0.23)^2\right)} = 154.1761.$$

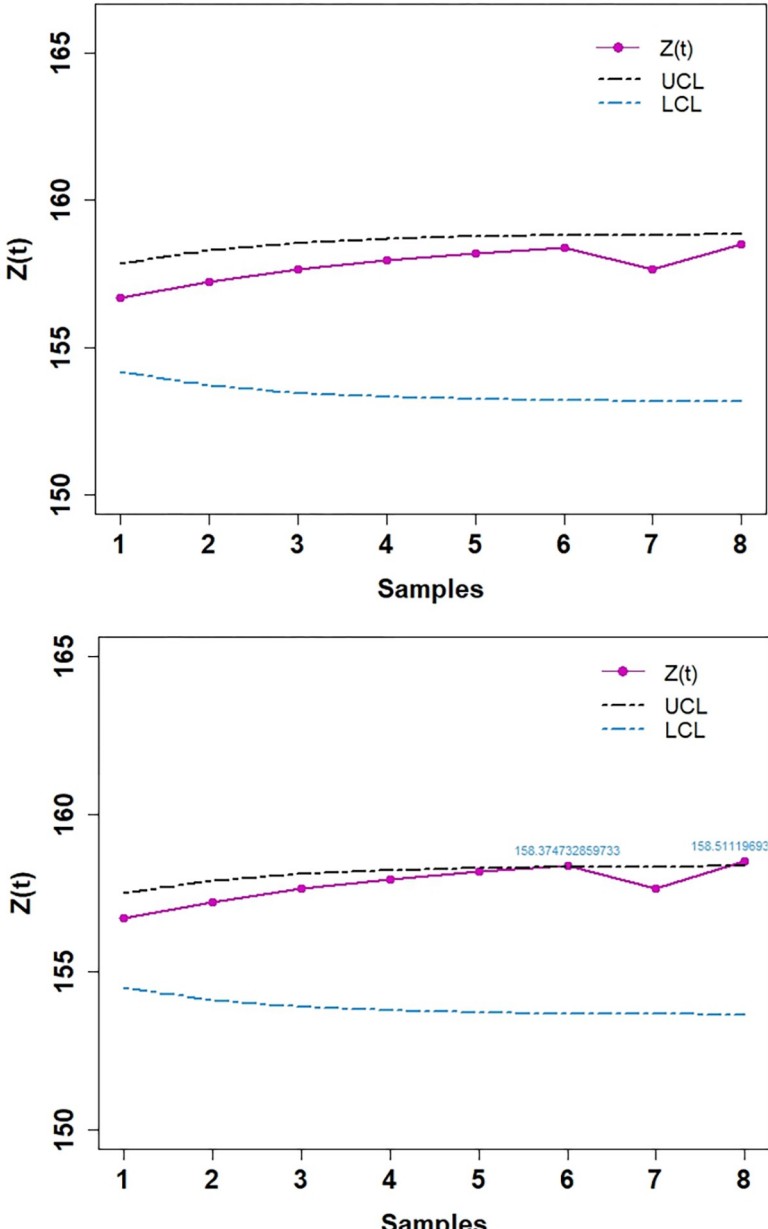

**Fig 18. EWMA CC for data set 2.** (a) Based on LSE. (b) Based on WLSE.

The second control limits of EWMA CC based on LSE from Eq (5) and Eq (7) is calculated as

$$UCL_{Z_2} = \alpha_0 + L\sqrt{Var(\hat{\alpha}_{LSE})\left(\frac{\lambda}{2-\lambda}\right)\left(1-(1-\lambda)^4\right)}.$$

$$= 156.00001 + 4.7\sqrt{(2.846888)\left(\frac{0.23}{2-0.23}\right)\left(1-(1-0.23)^4\right)} = 158.3020.$$

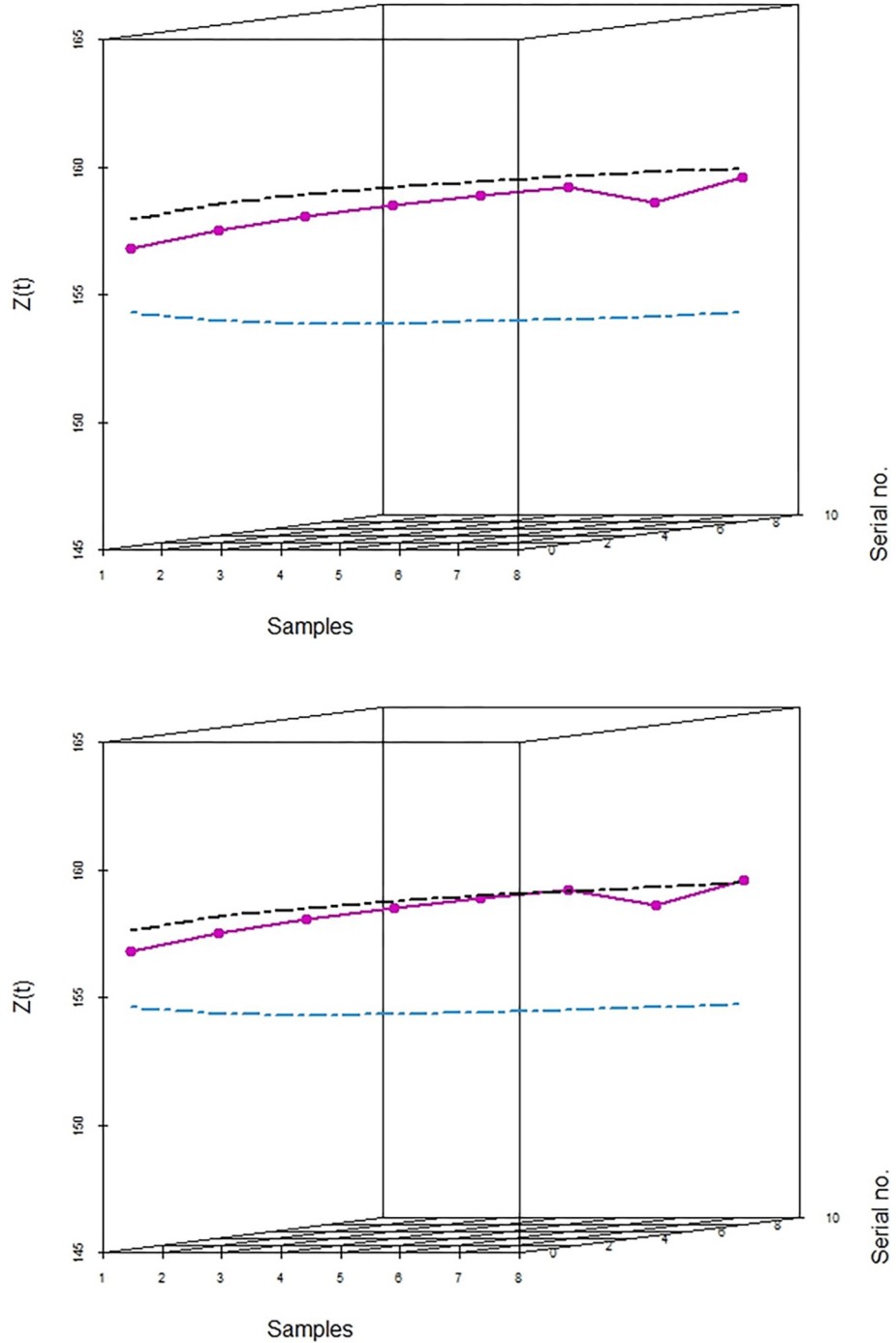

**Fig 19. 3D plot EWMA CC for data set 2.** (a) Based on LSE. (b) Based on WLSE.

$$LCL_{Z_2} = \alpha_0 - L\sqrt{Var(\hat{\alpha}_{LSE})\left(\frac{\lambda}{2-\lambda}\right)\left(1-(1-\lambda)^4\right)}.$$

$$= 156.00001 - 4.7\sqrt{(2.846888)\left(\frac{0.23}{2-0.23}\right)\left(1-(1-0.23)^4\right)} = 153.6980.$$

For EWMA CC based on WLSE, the first value of EWMA is calculated as

$$Z_1 = \lambda\,\hat{\alpha}_{WLSE(1)} + (1-\lambda)Z_0.$$

$$= (0.23)(159.0000) + (1-0.23)(156.00001) = 156.6900.$$

The second value of EWMA is calculated as

$$Z_2 = \lambda\,\hat{\alpha}_{WLSE(2)} + (1-\lambda)Z_1.$$

$$= (0.23)(159.0001) + (1-0.23)(146.0292) = 157.2214.$$

The first control limits of EWMA CC based on WLSE from Eq (8) and Eq (10) are calculated as

$$UCL_{Z_1} = \alpha_0 + L\sqrt{Var(\hat{\alpha}_{WLSE})\left(\frac{\lambda}{2-\lambda}\right)\left(1-(1-\lambda)^2\right)}.$$

$$= 156.00001 + 3.9\sqrt{(2.846739)\left(\frac{0.23}{2-0.23}\right)\left(1-(1-0.23)^2\right)} = 157.5134.$$

$$LCL_{Z_1} = \alpha_0 - L\sqrt{Var(\hat{\alpha}_{WLSE})\left(\frac{\lambda}{2-\lambda}\right)\left(1-(1-\lambda)^2\right)}.$$

$$= 156.00001 - 3.9\sqrt{(2.846739)\left(\frac{0.23}{2-0.23}\right)\left(1-(1-0.23)^2\right)} = 154.4866.$$

The second control limits of EWMA CC based on WLSE from Eq (8) and Eq (10) are calculated as

$$UCL_{Z_2} = \alpha_0 + L\sqrt{Var(\hat{\alpha}_{WLSE})\left(\frac{\lambda}{2-\lambda}\right)\left(1-(1-\lambda)^4\right)}.$$

$$= 156.00001 - 3.9\sqrt{(2.846739)\left(\frac{0.23}{2-0.23}\right)\left(1-(1-0.23)^4\right)} = 157.9101.$$

$$LCL_{Z_2} = \alpha_0 - L\sqrt{Var(\hat{\alpha}_{WLSE})\left(\frac{\lambda}{2-\lambda}\right)\left(1-(1-\lambda)^4\right)}.$$

$$= 156.00001 - 3.9\sqrt{(2.846739)\left(\frac{0.23}{2-0.23}\right)\left(1-(1-0.23)^4\right)} = 154.0899.$$

## 5. Proposed method can be adapted for different types of distributions

The different probability distributions have different parameters, for example: location, scale and shape parameters. It is not necessary that every distribution has every type of parameter. As in normal distribution there is no shape parameter. In this study, in both proposed control charts, we monitored the shape parameter of NKP distribution. It is the unique idea for the readers to adapt such proposed method for obtaining new control charts for number of symmetric or asymmetric distributions.

## 6. Future research directions

The readers not only can monitor the shape parameter but also can monitor the location, scale or some other parameters for introducing new EWMA control charts based on different distributions. It is also notable that this method not basically be used for EWMA control charts, but also can adapt for different control charts. For example, this method can adapt for double exponentially weighted average (DEWMA), hybrid exponentially weighted moving average control charts (HEWMA), and real applications of these new control charts can be reported for industry, agriculture and other different sectors.

## 7. Conclusion

In this study, new EWMA control charts are designed depending on a flexible model. These charts are based on the least square and weighted least square estimators of shape parameter of the new Kumaraswamy Pareto distribution. Both control charts are compared for checking their performance. The results are explored through numerical values and also with half a dozen plots. We examined the numerical results and the plots for the EWMA control chart based on weighted least square estimator, and we find that it has a better performance than the other proposed chart. Several key findings are reported which are obtained from the comparative analysis of EWMA control charts. The simulation study of proposed charts is also discussed in detail. The effectiveness of both proposed charts is examined through two real data sets. We mentioned that the proposed method can be adapted for different types of probability distributions. Some new future research directions are also recommended for the readers.

## Author Contributions

**Conceptualization:** Riffat Jabeen, Mashhood Ahmad, M. Nagy.

**Data curation:** Mashhood Ahmad.

**Formal analysis:** Riffat Jabeen, Hazem Al-Mofleh.

**Investigation:** Riffat Jabeen, Mashhood Ahmad, Azam Zaka, Hazem Al-Mofleh.

**Methodology:** Riffat Jabeen, Mashhood Ahmad, Azam Zaka, M. Nagy.

**Project administration:** Riffat Jabeen, Mashhood Ahmad.

**Resources:** Riffat Jabeen, Mashhood Ahmad, Azam Zaka, M. Nagy, Hazem Al-Mofleh.

**Software:** Riffat Jabeen, Hazem Al-Mofleh.

**Validation:** Azam Zaka, M. Nagy.

**Visualization:** Riffat Jabeen, Azam Zaka, Hazem Al-Mofleh.

**Writing – original draft:** Riffat Jabeen, Mashhood Ahmad, Azam Zaka.

**Writing – review & editing:** Riffat Jabeen, Mashhood Ahmad, Azam Zaka, M. Nagy, Hazem Al-Mofleh.

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
