## [Decision Letter · Decision Letter 0]

23 Jul 2024

PONE-D-24-21795A Novel Detection Method of Electronic Component Failure Time and Yield Strengths of Circular Tubes under Flexible ModelPLOS ONE

Dear Dr. Jabeen,

Thank you for submitting your manuscript to PLOS ONE. After careful consideration, we feel that it has merit but does not fully meet PLOS ONE’s publication criteria as it currently stands. Therefore, we invite you to submit a revised version of the manuscript that addresses the points raised during the review process.

We look forward to receiving your revised manuscript.

Kind regards,

M. Jagabar Sathik

Academic Editor

PLOS ONE

3. For studies involving third-party data, we encourage authors to share any data specific to their analyses that they can legally distribute. PLOS recognizes, however, that authors may be using third-party data they do not have the rights to share. When third-party data cannot be publicly shared, authors must provide all information necessary for interested researchers to apply to gain access to the data. (https://journals.plos.org/plosone/s/data-availability#loc-acceptable-data-access-restrictions) 

Additional Editor Comments:

Please address the comments and questions raised by the reviewers.

Reviewers' comments:

Reviewer's Responses to Questions

**Comments to the Author**

1. Is the manuscript technically sound, and do the data support the conclusions?

Reviewer #1: Yes

Reviewer #2: Yes

2. Has the statistical analysis been performed appropriately and rigorously? 

Reviewer #1: Yes

Reviewer #2: Yes

3. Have the authors made all data underlying the findings in their manuscript fully available?

Reviewer #1: Yes

Reviewer #2: Yes

4. Is the manuscript presented in an intelligible fashion and written in standard English?

Reviewer #1: Yes

Reviewer #2: Yes

5. Review Comments to the Author

Reviewer #1: Major comments:

1. The title of the paper is not according to the topic. I suggest the revision in the title.

2. I suggest not to define the abbreviations in the abstract. Also once an abbrivation is defined should not be defined again or should not be used in full for the next time.

3. The manuscript needs to a comprehensive improvement for its language and presentation.

4. Why the author used new Kumaraswamy-Pareto (NKP) distribution as the baseline distribution of the EWMA chart.Give the motivation behind this.

5. Add some latest research articles in introduction. The author may consider the following papers related to nonnormal cases: https://doi.org/10.1038/s41598-023-47159-9, https://doi.org/10.1038/s41598-024-52109-0.

6. Explain the motivation behind comparing LSE specifically with the WLSE.

7. Include some more graphical pepresentations of the results in the manuscript.

Minor comments:

1. The authors are suggested to Extend Table 1 by adding some more shifts.

2. Check formatting specially besides Figures and Tables

3. Figures should be interpreted in more details.

4. Conclusion is weak. It requires improvement.

5. References should write with same style.

Reviewer #2: This study presents a novel method for detecting electronic component failure time and yield strengths of circular tubes under a flexible model. It introduces new Exponential Weighted Moving Average (EWMA) control charts based on the least square and weighted least square estimators of the shape parameter of the new Kumaraswamy-Pareto (NKP) distribution, validated through simulation and real data sets. The proposed EWMA control charts offer a significant improvement over traditional methods by effectively monitoring the shape parameters of the NKP distribution, providing more accurate detection of out-of-control processes. The charts were shown to perform exceptionally well in simulations and real-life applications, demonstrating their practical utility in quality control for industrial processes.

I recommend to publish the paper after considering the following comments and corrections.

1. Define all acronyms at their first occurrence in the text.

2. Provide more detailed explanations for the terms "least square estimator (LSE)" and "weighted least square estimator (WLSE)" in the introduction.

3. Clarify the significance of the shape parameter in the context of the NKP distribution.

4. Include a brief overview of existing EWMA control charts to highlight the novelty of the proposed method.

5. Add references to support the claims made about the flexibility of the NKP distribution.

6. Ensure that all figures are properly labeled and referenced in the text.

7. Explain the choice of parameters (α, γ, θ) used in the simulation study.

8. Discuss the potential limitations of the proposed control charts.

9. Provide more context on the selection of the two real data sets used for validation.

10. Elaborate on the implications of the results obtained from the simulation study.

11. Include a table summarizing the key findings from the comparative analysis of EWMA control charts.

12. Ensure that the mathematical notation is consistent throughout the manuscript.

13. Clarify the role of the smoothing constant (λ) in the EWMA statistics.

14. Discuss how the proposed method can be adapted for different types of data distributions.

15. Include a section on future research directions based on the findings of this study.

16. Ensure that the abstract provides a concise summary of the methodology and key results.

17. Revisit the conclusion to emphasize the practical applications of the proposed control charts.

18. Provide a detailed description of the software and tools used for the simulation study.

19. Ensure that all references are cited correctly and are relevant to the study.

20. Review the manuscript for grammatical errors and improve the overall readability.

6. PLOS authors have the option to publish the peer review history of their article (what does this mean?). If published, this will include your full peer review and any attached files.

Reviewer #1: No

Reviewer #2: No

---

## [Author Response · Author response to Decision Letter 0]

17 Aug 2024

Our Reply to Reviewers

Before we begin answering the comments of the two reviewers point by point, we would like to thank them for important comments we think they have been paved the way towards a significant improvement of the paper. Now, we will present our reply for specific comments.

• The referee comments are shown in black.

• Our responses for reviewer #1 are shown in red, and responses for reviewer #2 are shown in blue.

• Furthermore, the corrections in the revised version based on the comments of reviewer #1 are shown in red, whereas the corrections in the revised version based on the comments of reviewer #2 are shown in blue.

Reviewer #1 Responses to Reviewer #1

Major comments: 

The title of the paper is not according to the topic. I suggest the revision in the title We modified the title of the article according to proposed work. 

I suggest not to define the abbreviations in the abstract. Also once an abbreviation is defined should not be defined again or should not be used in full for the next time. We excluded all abbreviation from abstract. We defined all abbreviations only one time.

The manuscript needs to a comprehensive improvement for its language and presentation. Thanks for your comment. The English language and presentation of the paper are improved throughout the paper.

Why the author used new Kumaraswamy-Pareto (NKP) distribution as the baseline distribution of the EWMA chart.Give the motivation behind this. As we can see this distribution is very flexible in its baseline article and can almost completely fitted on every data In that article it checked through goodness of fit measures that it is better fit as compare to various existing distributions Now anyone who wants to use our proposed charts in industry or agriculture sector, not only can applied our proposed control charts on any data with good results but also can show that this distribution is better fit, through goodness of fit measures, for this data.

Add some latest research articles in introduction. We added latest articles in introduction.

Explain the motivation behind comparing LSE specifically with the WLSE. The EWMA CC based on WLSE will be always better than EWMA CC based on LSE. There are two philosophies which run behind this argument. First philosophy is that WLSE is weighted form (modified form) of least square estimator. The second philosophy is that we have seen in its baseline article [1] that all estimators with all estimation methods, the WLSE is better than LSE.

Include some more graphical presentations of the results in the manuscript We included half dozen plots and some 3D plots also. 

Minor comments: 

The authors are suggested to Extend Table 1 by adding some more shifts. We added shift value and obtained new ARLs values.

Check formatting specially besides Figures and Tables We checked formatting specially besides Figures and Tables.

Figures should be interpreted in more details. Figures interpreted in more details.

Conclusion is weak. It requires improvement. Conclusion changed with improvement.

References should write with same style References wrote with same style.

Reviewer #2 Responses to Reviewer #2

Define all acronyms at their first occurrence in the text. We defined all acronyms at their first occurrence.

Provide more detailed explanations for the terms "least square estimator (LSE)" and "weighted least square estimator (WLSE)" in the introduction. We provided detailed explanation for terms "least square estimator (LSE)" and "weighted least square estimator (WLSE)" in the introduction.

Clarify the significance of the shape parameter in the context of the NKP distribution We clarified the significance of the shape parameter in the context of the NKP distribution

Include a brief overview of existing EWMA control charts to highlight the novelty of the proposed method. We wrote a brief overview of existing EWMA control charts to highlight the novelty of the proposed method.

Add references to support the claims made about the flexibility of the NKP distribution We added references to support the statements made about the NKP distribution.

Ensure that all figures are properly labeled and referenced in the text. All figures are properly labeled and referenced in the text.

Explain the choice of parameters (α, γ, θ) used in the simulation study We explained the choice of parameters (α, γ, θ) used in the simulation study.

Discuss the potential limitations of the proposed control charts. We made separate heading of limitations of the proposed control charts.

Provide more context on the selection of the two real data sets used for validation. We discussed two data sets in detail.

Elaborate on the implications of the results obtained from the simulation study. The discussed results obtained from the simulation study in detail.

Include a table summarizing the key findings from the comparative analysis of EWMA control charts We Included a table summarizing the key findings from the comparative analysis of EWMA control charts

Ensure that the mathematical notation is consistent throughout the manuscript We made the mathematical notations consistent throughout the manuscript.

Clarify the role of the smoothing constant (λ) in the EWMA statistics. We clarified the role of the smoothing constant (λ) in the EWMA statistics.

Discuss how the proposed method can be adapted for different types of data distributions. We made separate heading of the proposed method can be adapted for different types of data distributions.

Include a section on future research directions based on the findings of this study. We made heading of future research directions based on the findings of this study.

Ensure that the abstract provides a concise summary of the methodology and key results We changed almost completely abstract. Now it show a concise summary of the methodology and key results. 

Revisit the conclusion to emphasize the practical applications of the proposed control charts. We rewrote the conclusion to emphasize the practical applications of the proposed control charts.

Provide a detailed description of the software and tools used for the simulation study. We added detailed description of the software and various tools used for the simulation study.

Ensure that all references are cited correctly and are relevant to the study. We checked again references and tried to write it correctly.

Review the manuscript for grammatical errors and improve the overall readability. We review the manuscript for grammatical errors and improve the overall readability.

Please send the attached file ''Response to Reviewers'' to the reviewers. It is obvious than this window.

---

## [Decision Letter · Decision Letter 1]

26 Aug 2024

Unique Exploration Method of Electronic Component Failure Time and Yield Strengths of Circular Tubes Under Complete Flexible Model

PONE-D-24-21795R1

Dear Dr. Jabeen,

We’re pleased to inform you that your manuscript has been judged scientifically suitable for publication and will be formally accepted for publication once it meets all outstanding technical requirements.

Kind regards,

M. Jagabar Sathik

Academic Editor

PLOS ONE

Additional Editor Comments (optional):

The authors made necessary changes and the responses are satisfactory.

Reviewers' comments:

Reviewer's Responses to Questions

**Comments to the Author**

1. If the authors have adequately addressed your comments raised in a previous round of review and you feel that this manuscript is now acceptable for publication, you may indicate that here to bypass the “Comments to the Author” section, enter your conflict of interest statement in the “Confidential to Editor” section, and submit your "Accept" recommendation.

Reviewer #1: (No Response)

Reviewer #2: All comments have been addressed

2. Is the manuscript technically sound, and do the data support the conclusions?

Reviewer #1: (No Response)

Reviewer #2: Yes

3. Has the statistical analysis been performed appropriately and rigorously? 

Reviewer #1: (No Response)

Reviewer #2: Yes

4. Have the authors made all data underlying the findings in their manuscript fully available?

Reviewer #1: (No Response)

Reviewer #2: Yes

5. Is the manuscript presented in an intelligible fashion and written in standard English?

Reviewer #1: (No Response)

Reviewer #2: Yes

6. Review Comments to the Author

Reviewer #1: (No Response)

Reviewer #2: The authors have answered all of our comments and concerns. No further inquiries. I recommend to accept it in the current form

7. PLOS authors have the option to publish the peer review history of their article (what does this mean?). If published, this will include your full peer review and any attached files.

Reviewer #1: No

Reviewer #2: No

---

## [Editor Report · Acceptance letter]

3 Sep 2024

PONE-D-24-21795R1 

PLOS ONE

Dear Dr. Jabeen, 

I'm pleased to inform you that your manuscript has been deemed suitable for publication in PLOS ONE. Congratulations! Your manuscript is now being handed over to our production team.

Kind regards, 

on behalf of

Dr. M. Jagabar Sathik 

Academic Editor

PLOS ONE